# IL-17-producing γδ T cells switch migratory patterns between resting and activated states

Duncan R. McKenzie[1], Ervin E. Kara[1,†], Cameron R. Bastow[1], Timona S. Tyllis[1], Kevin A. Fenix[1], Carly E. Gregor[1], Jasmine J. Wilson[1], Rachelle Babb[1,†], James C. Paton[1,2], Axel Kallies[3,4], Stephen L. Nutt[3,4], Anne Brüstle[5], Matthias Mack[6], Iain Comerford[1,*] & Shaun R. McColl[1,7,*]

Interleukin 17-producing γδ T (γδT17) cells have unconventional trafficking characteristics, residing in mucocutaneous tissues but also homing into inflamed tissues via circulation. Despite being fundamental to γδT17-driven early protective immunity and exacerbation of autoimmunity and cancer, migratory cues controlling γδT17 cell positioning in barrier tissues and recruitment to inflammatory sites are still unclear. Here we show that γδT17 cells constitutively express chemokine receptors CCR6 and CCR2. While CCR6 recruits resting γδT17 cells to the dermis, CCR2 drives rapid γδT17 cell recruitment to inflamed tissues during autoimmunity, cancer and infection. Downregulation of CCR6 by IRF4 and BATF upon γδT17 activation is required for optimal recruitment of γδT17 cells to inflamed tissue by preventing their sequestration into uninflamed dermis. These findings establish a lymphocyte trafficking model whereby a hierarchy of homing signals is prioritized by dynamic receptor expression to drive both tissue surveillance and rapid recruitment of γδT17 cells to inflammatory lesions.

[1] Department of Molecular & Cellular Biology, University of Adelaide, Adelaide, South Australia 5005, Australia. [2] Research Centre for Infectious Diseases, University of Adelaide, Adelaide, South Australia 5005, Australia. [3] The Walter and Eliza Hall Institute of Medical Research, Parkville, Victoria 3052, Australia. [4] Department of Medical Biology, University of Melbourne, Parkville, Victoria 3010, Australia. [5] Department of Immunology and Infectious Diseases, John Curtin School of Medical Research, Australian National University, Canberra, Australian Capital Territory 2601, Australia. [6] Department of Internal Medicine II, University Hospital Regensburg, Regensburg 93042, Germany. [7] Centre for Molecular Pathology, University of Adelaide, Adelaide, South Australia 5005, Australia. * These authors contributed equally to this work. † Present address(es): Laboratory of Molecular Immunology, The Rockefeller University, New York, New York 10065, USA. (E.E.K.); Department of Medicine, Albert Einstein College of Medicine, New York, New York 10561, USA. (R.B.). Correspondence and requests for materials should be addressed to I.C. (email: iain.comerford@adelaide.edu.au) or to S.R.M. (email: shaun.mccoll@adelaide.edu.au).

Interleukin-17-producing γδ T cells (γδT17 cells) are innate-like lymphocytes crucial for early defence against extracellular bacterial and fungal pathogens. γδT17 effector function is programmed in Vγ4[+] and Vγ6[+] cells during thymic development, resulting in their homeostatic localization to barrier tissues and ability to be rapidly activated by innate-derived cytokines[1,2]. Production of interleukin 17A (IL-17A) and other inflammatory cytokines by γδT17 cells within hours of pathogen encounter orchestrates early neutrophil responses critical for mucocutaneous defence[3–5]. However, dysregulated γδT17 cell responses contribute to pathogenesis associated with several models of autoimmunity and can enhance tumour growth and metastasis[1,6–9].

How γδT17 cells populate homeostatic barrier tissues and then infiltrate inflamed tissues from circulation is unclear. γδT17 cells seed dermis and mucosal tissues during perinatal life[10]. Although parabiosis experiments demonstrate that the majority of Vγ4[+] γδT17 cells in skin-draining lymph nodes (sLNs) are permanently resident[11], studies using photolabelling, adoptive transfers and receptor antagonism suggest that γδT17 cells constitutively circulate between dermis, sLNs and blood[10,12–14]. Nevertheless, sLN γδT17 cells expand during autoimmune inflammation and infiltrate target tissues via circulation[1,9]. Furthermore, dermal Vγ4[+] γδT17 cells home from skin to sLNs, proliferate, and repopulate inflamed and distal unaffected skin during psoriasis[15]. Thus despite a largely tissue-restricted distribution, γδT17 cells are motile and move between lymphoid and barrier tissues under homeostasis and experimental inflammatory conditions.

Chemokine receptor CCR6, involved in both homeostatic and inflammatory trafficking of leukocytes in barrier tissues, is expressed by both T helper 17 (Th17) and γδT17 cells[16,17]. We reported a largely redundant function for CCR6 in recruitment of granulocyte–macrophage colony stimulating factor-producing encephalitogenic Th17 cells to the central nervous system (CNS) during experimental autoimmune encephalomyelitis (EAE). Instead, these cells display a CCR6[−]CCR2[+] phenotype and infiltrate the CNS via CCR2, which is critical for T-cell-driven pathology[18]. In γδT17 cell biology, CCR6 has a debated function in regulating Vγ4[+] cell homeostasis, and is reported to direct γδT17 cell trafficking during inflammation[10,11,19]. However, Vγ4[+] cells homing from inflamed skin to sLNs during psoriasis predominantly lack CCR6 expression[14]. By contrast, CCR2 is implicated in the migration of γδT17 cells to psoriatic skin and arthritic synovium[15,20], pointing to a clear interplay between CCR6 and CCR2 function in control of γδT17 cell homing. Nevertheless, a clear understanding of γδT17 cell trafficking mechanisms at rest and during inflammation is lacking.

Here, we find that CCR6 controls homeostatic γδT17 cell trafficking to the dermis, whereas constitutive CCR2 expression drives their rapid homing to inflammatory sites. In models of autoimmunity, cancer and infection, activation-induced down-regulation of CCR6 releases γδT17 cells from their homeostatic immunosurveillance trafficking circuit through the skin and circulation, which then enhances their CCR2-dependent homing to inflamed tissue. Therefore, the dynamic interplay between CCR6 and CCR2 expression defines γδT17 cell trafficking patterns between resting and activated states.

## Results

**γδT17 cells downregulate CCR6 upon activation.** We recently reported that Th17 cell development during EAE is coupled with a dynamic, temporally regulated switch from CCR6 to CCR2 expression as Th17 cells propagate their differentiation. Expression patterns of CCR6 and CCR2 define distinct effector phenotypes of Th17 cells, with a CCR6[−]CCR2[+] phenotype marking the encephalitogenic granulocyte–macrophage colony-stimulating factor/interferon-γ-producing population[18]. Unlike Th17 cells, γδT17 cell effector function is programmed during thymic development and these cells populate barrier tissues prior to inflammation[2,21,22]. Thus, we initially examined CCR6 and CCR2 expression in sLN and dermis in unimmunized *Il17a^Cre × Rosa26^eYFP* mice, where *Il17a* expression drives permanent marking of cells with eYFP[23]. γδT17 cells in these compartments constitutively co-expressed CCR2 and CCR6 (Fig. 1a and Supplementary Fig. 1a). Expression of CCR6 and CCR2 was restricted to γδ T cells bearing a CD27[−]CD44[hi] phenotype, characteristic of γδT17 cells (Supplementary Fig. 2a)[24]. CCR6/CCR2 co-expression was similar between Vγ4[+] and Vγ6[+] γδT17 cell subsets as distinguished by both Vγ4 expression and CD3/T-cell receptor (TCR) expression level, as previously reported ('CD3[bright] staining')[25] (Supplementary Fig. 1b,c), and both receptors were functional as determined by *ex vivo* chemotaxis (Fig. 1b). However, examination of γδT17 cells from diverse tissues revealed a heterogeneous pattern of CCR6 expression. While thymic and most lymphoid γδT17 cells uniformly expressed both CCR6 and CCR2, populations of γδT17 cells lacking CCR6 expression (CCR6[−]CCR2[+]) were prominent in lung and gut-associated tissues (Fig. 1c). As the gut is tonically immunologically active due to interactions with commensal microbiota, we hypothesized that γδT17 cells downregulate CCR6 expression during inflammation.

In support of this idea, activation of sLN and spleen γδT17 cells *in vivo* during EAE resulted in downregulation of CCR6 expression compared to unimmunized mice (Fig. 1d). CNS-infiltrating γδT17 cells were also largely CCR6[−]. BrdU incorporation revealed that CCR6 expression was downregulated in proliferated γδT17 cells, while BrdU[−] cells remained CCR6[+] (Fig. 1e). Other γδ T-cell subsets did not express CCR2 or CCR6 at rest, and did not gain expression of these receptors over the course of EAE (Supplementary Fig. 2a). Unlike Th17 cells, γδT17 cells are predominantly activated by TCR-independent signals including IL-23 and IL-1β[1]. *In vitro* stimulation of lymphocytes with a range of known stimuli including IL-23/IL-1β, IL-23/IL-18 (ref. 26), IL-7 (ref. 27) and γδ-TCR signalling[28] uniformly repressed CCR6 surface expression in γδT17 cells (Fig. 1f and Supplementary Fig. 2b). IL-12 did not impact CCR6 expression, consistent with a reported absence of IL-12R expression by γδT17 cells[29] (Supplementary Fig. 2b). Activation-induced CCR6 downregulation correlated with induction of activation markers CD69 and CD25 and increased CD44 expression, and occurred in both Vγ4[+] and Vγ6[+] γδT17 cells (Supplementary Fig. 2c,d). In all *in vivo* and *in vitro* systems, γδT17 cells maintained high levels of CCR2 following activation, and virtually all γδT17 cells were CCR2[+] (Fig. 1a,c,d,f). Therefore, γδT17 cells are programmed to co-express CCR6 and CCR2 during development, but lose CCR6 expression upon activation.

**CCR2 drives γδT17 cell recruitment to inflamed tissues.** Tissue-infiltrating γδT17 cells are best understood in the context of cancer and autoimmunity. γδT17 cells infiltrate B16 melanomas and promote tumour growth[30,31], and infiltrate the CNS at disease onset and exacerbate disease pathogenesis during EAE[1,32]. How γδT17 cells infiltrate these inflammatory lesions is unknown. We thus used these models to investigate CCR6 and CCR2 function in control of γδT17 cell migration during inflammation. Consistent with the observation that activation induces downmodulation of CCR6 expression, *Ccr6*-deficiency did not affect γδT17 cell infiltration of B16 melanomas (Fig. 2a,b), nor recruitment to the CNS during EAE onset

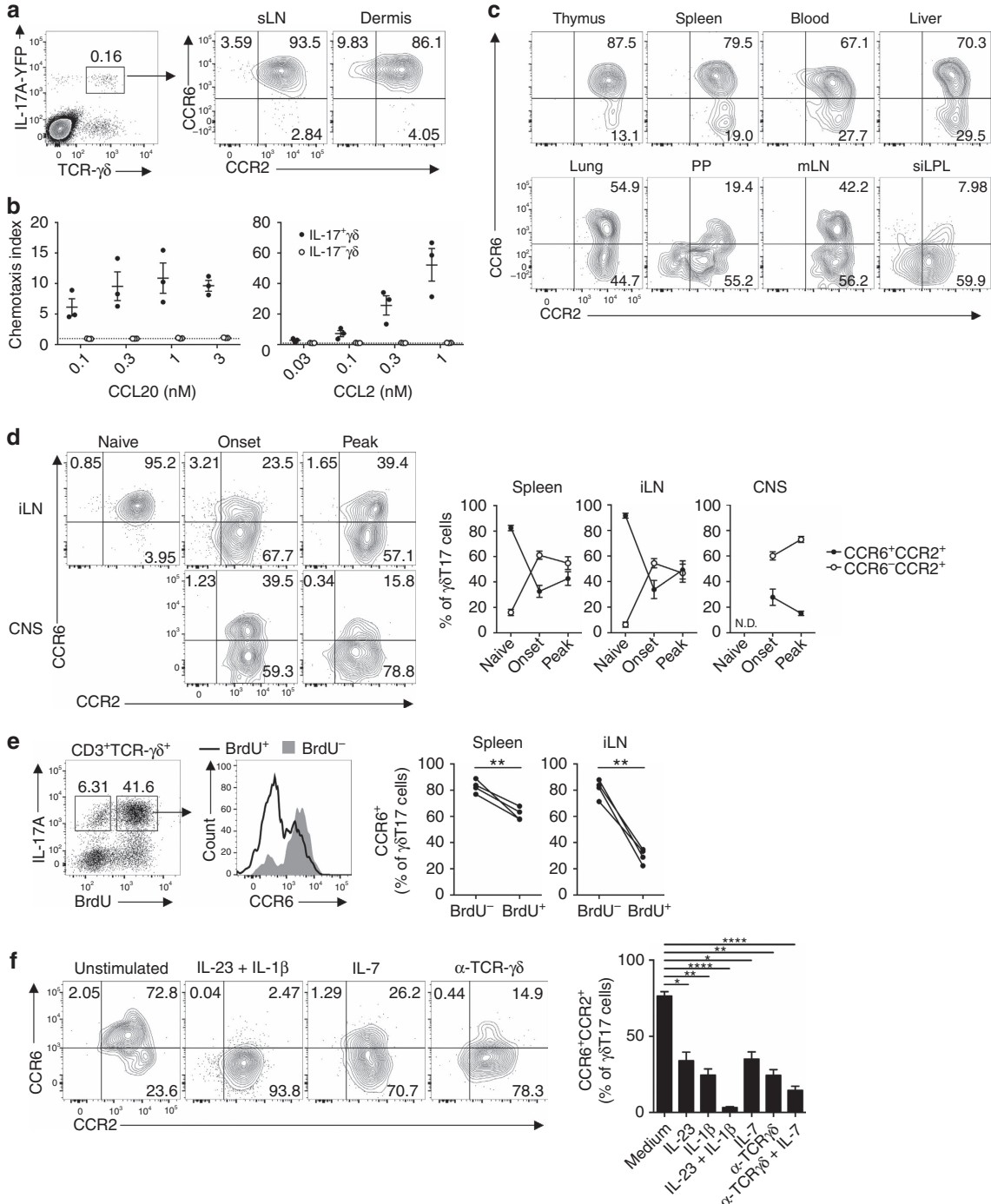

**Figure 1 | γδT17 cells downregulate CCR6 upon activation.** (**a**) Representative flow cytometry of CCR6 and CCR2 expression in skin-draining lymph nodes (sLN) and dermal CD3$^+$TCR-γδ$^+$IL-17A-YFP$^+$ γδT17 cells from $Il17a^{Cre} \times Rosa26^{eYFP}$ mice ($n = 3$). (**b**) $Ex\ vivo$ transwell chemotaxis of $Il17a^{Cre} \times Rosa26^{eYFP}$ splenic IL-17A$^{+/-}$ γδ T cells to CCL20 and CCL2 ($n = 3$). (**c**) Representative flow cytometry of CD45$^+$ γδT17 cells from organs of naïve $Il17a^{Cre} \times Rosa26^{eYFP}$ mice ($n = 3$). mLN, mesenteric lymph node; PP, Peyer's patches; siLPL, small intestinal lamina propria lymphocytes. (**d**) Representative flow cytometry and quantitation of CCR6 and CCR2 expression by γδT17 cells from organs of $Il17a^{Cre} \times Rosa26^{eYFP}$ mice either naïve ($n = 6$) or at experimental autoimmune encephalomyelitis (EAE) onset ($n = 7$) or peak ($n = 5$). CNS, central nervous system; iLN, inguinal lymph node; ND, not detected. (**e**) Representative flow cytometry and quantitation of CCR6 expression by γδT17 cells from wild type (WT) mice given BrdU at d3 post-immunization for EAE, and analysed at d8 ($n = 4$). (**f**) Representative flow cytometry and frequency of CCR6 and CCR2 expression by γδT17 cells from $Il17a^{Cre} \times Rosa26^{eYFP}$ lymphocytes cultured with indicated stimuli for 72 h ($n = 5$). See also Supplementary Figs 1 and 2. Mean ± s.e.m. (**a–c**) Representative of two experiments. (**d,f**) Pooled from two experiments. (**e**) Paired two–tailed Student's $t$-test, (**f**) one-way paired ANOVA with Dunnett's multiple comparisons test relative to unstimulated control. *$P < 0.05$, **$P < 0.01$, ***$P < 0.001$, ****$P < 0.0001$.

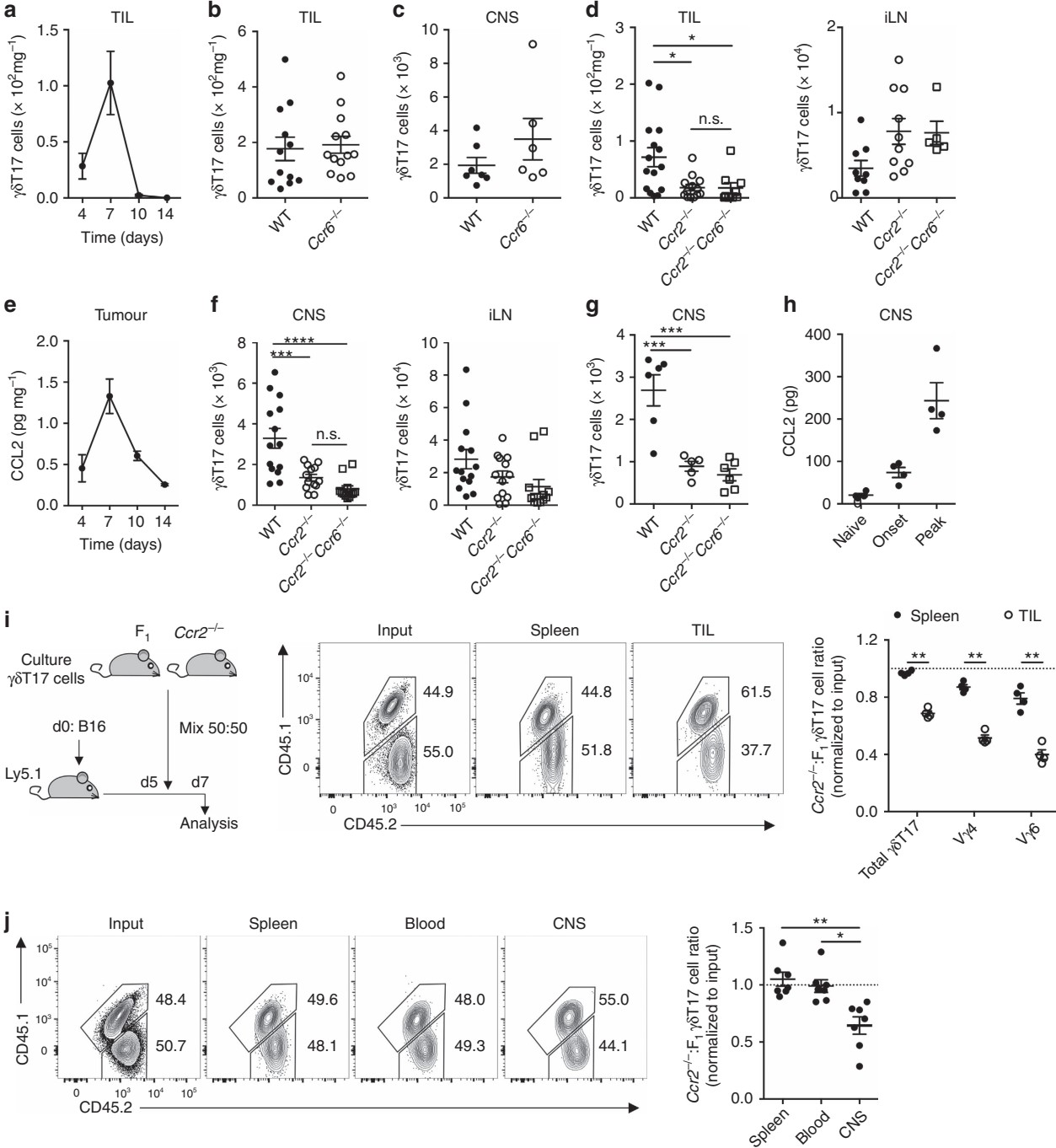

**Figure 2 | CCR2 recruits γδT17 cells to inflammatory sites.** (**a**) CD45$^+$CD3$^+$TCRγδ$^+$IL-17A$^+$ γδT17 cell numbers in tumour-infiltrating lymphocytes (TIL) following B16 melanoma challenge ($n=5$/time point). (**b**) γδT17 cell numbers in TIL d7 post-challenge with B16 melanoma in wild type (WT) ($n=12$) and $Ccr6^{-/-}$ mice ($n=13$). (**c**) γδT17 cell numbers in central nervous system (CNS) at experimental autoimmune encephalomyelitis (EAE) onset in WT ($n=7$) and $Ccr6^{-/-}$ mice ($n=6$). (**d**) γδT17 cell numbers in TIL and inguinal lymph nodes (iLN) d7 post-challenge with B16 melanoma in WT ($n=15$ (TIL), 9 (iLN)), $Ccr2^{-/-}$ ($n=13$ (TIL), 10 (iLN)) and $Ccr2^{-/-}Ccr6^{-/-}$ mice ($n=9$ (TIL), 5 (iLN)). (**e**) ELISA for CCL2 in tumour supernatant from WT mice challenged with B16 melanoma ($n=5$/time point). (**f**) γδT17 cell numbers in CNS and iLN at EAE onset in WT ($n=14$), $Ccr2^{-/-}$ ($n=13$) and $Ccr2^{-/-}Ccr6^{-/-}$ mice ($n=12$). (**g**) γδT17 cell numbers in CNS at peak disease in WT ($n=6$), $Ccr2^{-/-}$ ($n=5$) and $Ccr2^{-/-}Ccr6^{-/-}$ mice ($n=6$). (**h**) ELISA for CCL2 in CNS of WT mice with EAE ($n=4$/time point). (**i**) Ly5.1 mice ($n=4$) at d5 post-challenge with B16 melanoma were transferred i.v. with *in vitro*-expanded γδT17 cells from $Ccr2^{-/-}$ (CD45.2$^+$) and F$_1$ (CD45.1$^+$CD45.2$^+$) mice. $Ccr2^{-/-}$:F$_1$ total, Vγ4 and Vγ6 γδT17 cell ratios in spleen and tumours were normalized to input ratio. Vγ4 and Vγ6 γδT17 cells were determined by CD3$^{bright}$ gating. Representative flow cytometry of CD45.2$^+$ γδT17 cells at d7 or input. (**j**) Ly5.1 mice ($n=7$) at EAE onset were transferred with F$_1$ and $Ccr2^{-/-}$ γδT17 cells as in (**i**). Twenty-four hours later, ratios of $Ccr2^{-/-}$:F$_1$ γδT17 cells in spleen, blood and CNS were normalized to input. Representative flow cytometry of CD45.2$^+$ γδT17 cells 24 h later or input. See also Supplementary Figs 3 and 4. Mean ± s.e.m. (**a,c,e,i**) Representative of two experiments. (**b,d,f,j**) Pooled from two experiments. (**b,c**) Unpaired two-tailed Student's *t*-test, (**d,f–g,j**) one-way ANOVA with Bonferroni's multiple comparisons test (paired in **j**)), (**i**) paired two-tailed Student's *t*-test. *$P<0.05$, **$P<0.01$, ***$P<0.001$, ****$P<0.0001$.

(Fig. 2c). Thus, γδT17 cell trafficking to inflamed tissues in these settings occurs independently of CCR6.

In contrast, deficiency of *Ccr2* abrogated γδT17 cell infiltration of B16 melanomas but did not affect their expansion in draining lymph nodes (LNs) (Fig. 2d). CCR2-driven infiltration of γδT17 cells was consistent with upregulation of its major ligand CCL2 in tumours (Fig. 2e). Similar results were found in EAE, where *Ccr2*-deficiency inhibited γδT17 cell recruitment to the CNS at both onset and peak disease (Fig. 2f,g), time points at which CCL2 was induced in the CNS as reported[33] (Fig. 2h). CCR2 appeared to operate independently of CCR6 in regulation of γδT17 cell trafficking, as compound deficiency of *Ccr6* and *Ccr2* ($Ccr6^{-/-}Ccr2^{-/-}$) did not further affect γδT17 cell infiltration in either model (Fig. 2d,f,g).

$Ccr2^{-/-}Ccr6^{-/-}$ mice exhibited enhanced tumour growth, while $Ccr2^{-/-}$ and $Ccr2^{-/-}Ccr6^{-/-}$ mice had decreased EAE severity (Supplementary Fig. 3). Therefore, to examine the cell-intrinsic requirements of CCR2 for γδT17 cell migration, we developed a novel *in vitro* expansion protocol to generate large numbers of purified activated γδT17 cells, which contained both $V\gamma4^+$ and $V\gamma6^+$ subsets and maintained functional CCR2 expression (Supplementary Fig. 4). Equal ratios of *in vitro*-expanded genetically marked wild type (WT) and $Ccr2^{-/-}$ γδT17 cells were co-transferred into B16 melanoma-bearing recipients. While donor γδT17 cells recovered from spleen retained the input ratio, $Ccr2^{-/-}$ γδT17 cells did not migrate efficiently to tumours. This observation was true for both $V\gamma4^+$ and $V\gamma6^+$ subsets (Fig. 2i). Similar experiments using the EAE model revealed equivalent WT:$Ccr2^{-/-}$ γδT17 cell ratios in spleen and blood but reduced $Ccr2^{-/-}$ γδT17 cell recruitment to the CNS (Fig. 2j). Thus, CCR2, but not CCR6, drives activated γδT17 cell migration to inflammatory sites during B16 melanoma and EAE.

**CCR2 is essential for protective γδT17 cell responses**. The above models involve γδT17 cell infiltration from circulation, as the CNS and tumours lack resident γδT17 cell populations. However, many inflammatory scenarios implicate tissue-resident γδT17 cells, which survey and rapidly defend against infection at barrier surfaces. The extent to which γδT17 cell migration contributes to host defence during ongoing inflammation is unknown. To investigate whether CCR2 also directs tissue-infiltrating γδT17 cells during infection, we used experimental *Streptococcus pneumoniae* infection, immunity to which requires γδT17 cells[4]. Accordingly, $Tcrd^{-/-}$ mice had higher bacterial burden and reduced neutrophils in the nasal wash (NW) than WT at 72 h post-infection (Fig. 3a,b). *S. pneumoniae* infection induced γδT17 cell expansion in draining LNs and nasal-associated lymphoid tissue (Fig. 3c). CCL2 was induced in the nasal passages (NPs) upon infection (Fig. 3d), and co-transfer of *in vitro*-expanded WT and $Ccr2^{-/-}$ γδT17 cells into infected mice revealed an intrinsic requirement of CCR2 for γδT17 cell accumulation in NP (Fig. 3e). Thus, CCR2 drives circulating γδT17 cell infiltration of mucosal tissue during *S. pneumoniae* infection.

To elucidate the ability of recruited γδT17 cells to control infection, we transferred purified *in vitro*-expanded γδT17 cells into $Tcrd^{-/-}$ hosts prior to *S. pneumoniae* infection. In this model, tissue-infiltrating γδT17 cells provide the only source of γδ T-cell-driven protection. Transfer of WT γδT17 cells reduced nasopharyngeal bacterial burden by ∼10-fold, whereas $Ccr2^{-/-}$ γδT17 cells completely failed to control infection (Fig. 3f). Hence, CCR2 drives recruitment of protective γδT17 cells to the nasal mucosa during *S. pneumoniae* infection. Collectively, we conclude that γδT17 cell trafficking to diverse inflamed tissues is critically dependent on CCR2 signalling.

**CCR6 regulates homeostatic positioning of γδT17 cells**. Expression of CCR6 during γδT17 cell thymic development followed by rapid downregulation upon activation suggests that CCR6 plays a more prominent function in regulation of γδT17 cell homeostasis. While CCL20 is induced during inflammation, it is constitutively expressed in barrier tissues including skin, Peyer's patches and large intestine[34–37]. Both $Ccr6^{-/-}$ and $Ccr2^{-/-}Ccr6^{-/-}$ mice had markedly reduced number and frequency of γδ T cells expressing intermediate amounts of CD3/TCR in the dermis (γδT$^{lo}$), a population previously reported to produce IL-17 and distinct from TCR$^{hi}$ dendritic epidermal T cells[22] (Fig. 4a). We confirmed that γδT$^{lo}$ cells were entirely marked by eYFP in $Il17a^{Cre} \times Rosa26^{eYFP}$ mice, despite negligible IL-17A production following *ex vivo* restimulation (Supplementary Fig. 5a). In contrast to a previous report[10], *Ccr6*-deficiency reduced the number of both $V\gamma4^+$ and $V\gamma4^-$ ($V\gamma6^+$) γδT$^{lo}$ cells, although the ratio was skewed slightly towards $V\gamma6^+$ cells (Fig. 4b). Examination of other organs revealed that deficiency in *Ccr2* had no effect on γδT17 cell homeostasis, while *Ccr6*-deficiency increased γδT17 cells in the peritoneal cavity (Supplementary Fig. 5b). We conclude that CCR6 regulates dermal γδT17 cell residence.

To determine whether CCR6 drives recruitment of circulating γδT17 cells into dermis, we transferred unstimulated WT or $Ccr6^{-/-}$ lymphocytes into naïve mice and tracked their accumulation in dermis. Transferred WT γδT17 cells were substantially enriched in dermis, demonstrating that γδT17 cells can constitutively populate the skin from circulation. In contrast, $Ccr6^{-/-}$ γδT17 cells were defective in infiltration of dermis and pooled in the blood (Fig. 4c). In support of earlier results, both $V\gamma4^+$ and $V\gamma4^-$ γδT17 cells were recruited to the dermis, the ratio of which was unaltered by *Ccr6* deficiency, suggesting both populations are dependent on CCR6 for circulation-to-dermis trafficking (Fig. 4c). While constitutive expression of CCL20 in epidermis was reported, whether it is expressed in uninflamed dermis is unclear[34,38]. We found that *Ccl20* was constitutively expressed in the dermis by an uncharacterized $CD31^-$ $CD90^-$ $CD140\alpha^-$ stromal population (Fig. 4d). Thus, CCR6 directs homeostatic recruitment of γδT17 cells from circulation into dermis.

**IRF4 and BATF regulate CCR6 expression in γδT17 cells**. The downregulation of CCR6 upon γδT17 cell activation is surprising, as T cells typically upregulate inflammatory chemokine receptors upon activation. Consequently we investigated the underlying mechanism regulating this process. *Ccr6* transcript levels were reduced by approx. fourfold in γδT17 cells within 24 h of stimulation, whereas *Ccr2* expression was maintained (Fig. 5a and Supplementary Fig. 6a). This indicated that CCR6 expression is transcriptionally regulated during γδT17 cell activation. We thus examined expression of transcription factors previously implicated directly or indirectly in control of *Ccr6* expression, including RORγt[17], IRF4 (ref. 39), IRF8 (refs 40,41), Blimp1 (refs 39,42), BATF[43] and T-bet and Eomes[18]. *Rorc* (RORγt) was highly expressed in resting γδT17 cells but was downregulated by 24 h of activation. *Batf* and *Prdm1* (Blimp1) were rapidly upregulated by 24 h, while *Irf8* and *Irf4* were upregulated by 48 h, although *Irf4* was already present in resting γδT17 cells. Expression of *Eomes* and *Tbx21* (T-bet) at rest or following activation was minimal (Fig. 5b and Supplementary Fig. 6b). Therefore, we tested whether RORγt, IRF4, BATF, Blimp1 or IRF8 repressed *Ccr6* expression during γδT17 cell activation.

The similar expression kinetics and known *Ccr6* regulatory activity of RORγt presented the possibility that its downregulation may result in loss of CCR6 expression. To test this,

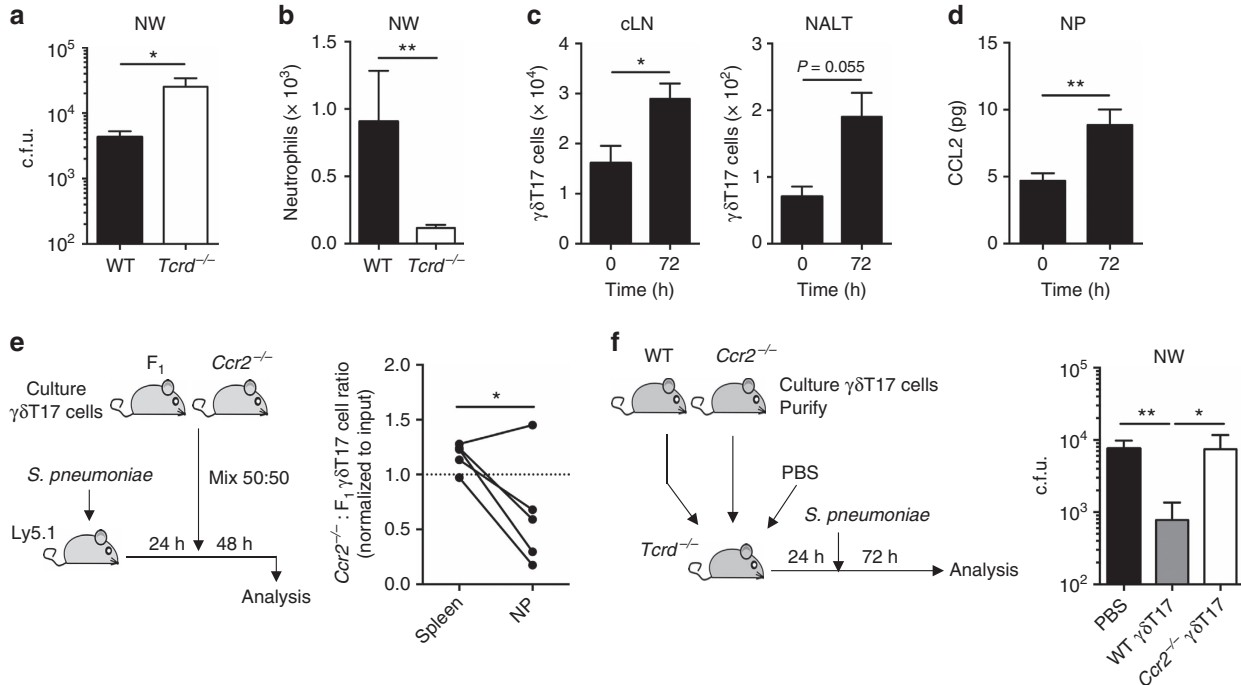

**Figure 3 | CCR2 drives protective γδT17 cell responses. (a)** Colony-forming units (c.f.u.) and **(b)** CD45⁺CD11b⁺Ly6G⁺ neutrophils recovered from nasal wash (NW) of wild type (WT) ($n=9$) and $Tcrd^{-/-}$ mice ($n=10$) 72 h post-infection with *S. pneumoniae*. **(c)** γδT17 cell numbers in cervical lymph node (cLN) and nasal-associated lymphoid tissue (NALT) and **(d)** ELISA for CCL2 in digested nasal passage (NP) supernatant in unimmunized mice ($n=7$) and at 72 h post-*S. pneumoniae* infection ($n=13$). **(e)** Ly5.1 mice ($n=5$) 24 h post-*S. pneumoniae* infection were transferred i.v. with expanded γδT17 cells from $Ccr2^{-/-}$ (CD45.2⁺) and F₁ (CD45.1⁺CD45.2⁺) mice. The $Ccr2^{-/-}$:F₁ γδT17 cell ratio in spleen and NP was normalized to input ratio. **(f)** Twenty-four hours prior to *S. pneumoniae* infection, $Tcrd^{-/-}$ hosts received PBS ($n=8$) or expanded and purified γδT17 cells from WT ($n=9$) or $Ccr2^{-/-}$ ($n=7$) mice. c.f.u. recovered from NW 72 h post-infection. Mean ± s.e.m. **(a–d)** Pooled from two experiments. **(a,b)** Mann–Whitney test, **(c,d)** unpaired two-tailed Student's *t*-test, **(e)** paired two-tailed Student's *t*-test, **(f)** Kruskal–Wallis test with Dunn's multiple comparisons test. *$P<0.05$, **$P<0.01$.

we retrovirally forced *Rorc* expression in *in vitro*-expanded γδT17 cells (Fig. 5c). However, this failed to alter CCR6 expression even in the highest GFP-expressing cells, suggesting that RORγt downregulation is not required for repression of CCR6 in activated γδT17 cells (Fig. 5d).

To determine whether IRF4, BATF, IRF8 or Blimp1 actively repress *Ccr6* expression, we cultured genetically-marked WT and transcription factor-deficient splenocytes with IL-23 and IL-1β. γδT17 cells were present in all strains although at differing frequencies, and homeostatic CCR6 expression was comparable to WT (Supplementary Fig. 6c). IRF4- and BATF-deficient γδT17 cells were intrinsically defective in both proliferation and CCR6 downregulation upon stimulation (Fig. 5e). IRF8 and Blimp1 were not required for these processes, although Blimp1 appeared to moderately promote CCR2 expression in activated γδT17 cells (Supplementary Fig. 6d). $Irf4^{-/-}$ and $Batf^{-/-}$ γδT17 cells exhibited comparable surface expression of IL-23R and IL-1R1 to WT cells, indicating that maintained CCR6 expression was likely due to defective signalling downstream of IL-23 and IL-1β stimulation (Supplementary Fig. 7a). Analysis of our and others' ChIP-Seq datasets in T cells[43–45] revealed binding of IRF4 and BATF to a shared site in the *Ccr6* promoter (Supplementary Fig. 7b), suggesting that these factors cooperatively and directly repress *Ccr6* in γδT17 cells. To assess whether defective proliferation in absence of IRF4 or BATF was the cause of impaired CCR6 downregulation, dye-labelled WT splenocytes were pre-treated with proliferation inhibitor mitomycin C prior to stimulation. Although proliferation was effectively blocked, CCR6 downregulation still occurred upon mitomycin C treatment, suggesting that proliferation and CCR6

downregulation are coincident but independent (Fig. 5f). Together, these data indicate that activation-induced CCR6 downregulation in γδT17 cells is promoted by IRF4 and BATF, and is largely uncoupled from proliferation.

**Loss of CCR6 promotes γδT17 cell homing to inflamed tissues.** Given the constitutive expression of CCL20 in mucocutaneous sites, we hypothesized that repression of CCR6 during activation enables homing of γδT17 cells toward inflammatory lesions by preventing their accumulation in uninflamed skin. To test this, we first compared the trafficking of *in vitro*-activated WT γδT17 cells with resting WT and $Ccr6^{-/-}$ γδT17 cells upon transfer into unimmunized hosts. Activated WT γδT17 cells demonstrated the same defect in homing to the dermis as resting $Ccr6^{-/-}$ γδT17 cells, and both pooled in blood compared to resting WT γδT17 cells (Fig. 6a). γδT17 cells lack CD62L and CCR7 expression, and traffic from skin to sLNs in a CCR7-independent manner[12]. Thus γδT17 cell entry to sLNs following adoptive transfer likely occurs via afferent lymph draining from dermis. In keeping with this idea, resting $Ccr6^{-/-}$ or *in vitro*-activated WT γδT17 cells, impaired in their ability to home to uninflamed skin, also accumulated less than resting WT γδT17 cells in sLNs (Fig. 6a). These data are consistent with the notion that activation switches off γδT17 cell homeostatic circulation patterns, enabling directed migration toward inflammatory cues.

To investigate this proposal directly, we studied the migratory patterns of *in vitro*-activated γδT17 cells retrovirally forced to maintain CCR6 expression. Infection with $Ccr6^{tg}$ virus restored CCR6 expression in activated γδT17 cells, which regained the

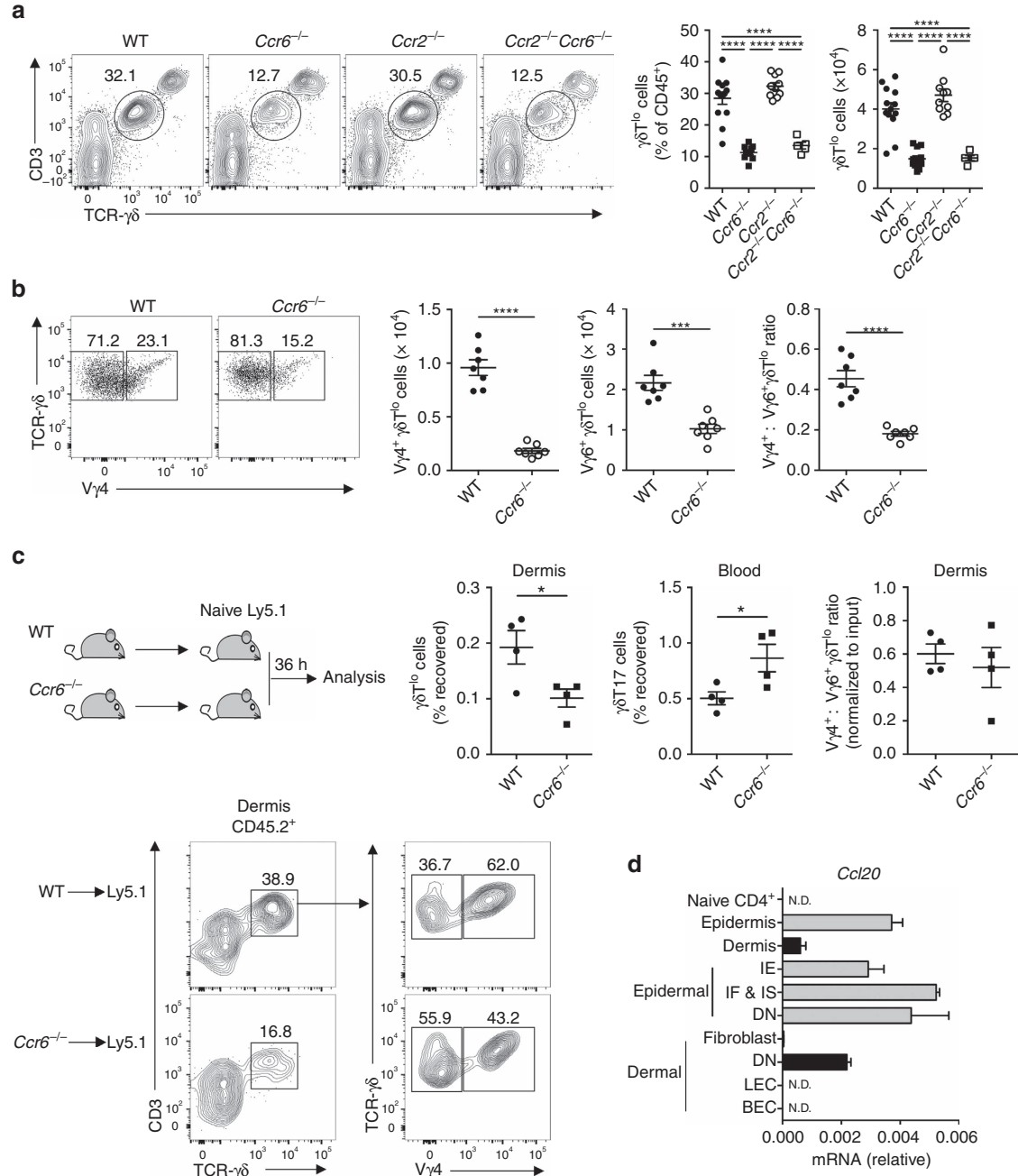

**Figure 4 | CCR6 regulates homeostatic γδT17 cell recruitment to dermis. (a)** Representative flow cytometry and quantitation of CD45$^+$CD3$^{lo}$TCR-γδ$^{lo}$ (γδT$^{lo}$) cells from ear skin dermis of naïve wild type (WT) ($n = 13$), $Ccr6^{-/-}$ ($n = 11$), $Ccr2^{-/-}$ ($n = 10$) and $Ccr2^{-/-}Ccr6^{-/-}$ mice ($n = 5$). **(b)** Representative flow cytometry of Vγ4 expression by dermal γδT$^{lo}$ cells and quantitation of Vγ4$^+$ and Vγ4$^-$ γδT$^{lo}$ cells in dermis of WT and $Ccr6^{-/-}$ mice ($n = 7$/group). **(c)** WT or $Ccr6^{-/-}$ lymphocytes were transferred i.v. into naïve Ly5.1 mice ($n = 4$/group). After 36 h, number of CD45.2$^+$ γδT$^{lo}$/γδT17 cells recovered was expressed as % of number transferred. Representative flow cytometry of dermal CD45.2$^+$ cells and quantitation of γδT17 cell recovery and Vγ4$^+$:Vγ4$^-$ ratio (normalized to input). **(d)** *Ccl20* mRNA from whole tissues or sorted CD45$^-$ epidermal keratinoctyes (Sca-1$^+$Ep-CAM$^{lo}$ interfollicular epidermis (IE), Sca-1$^{lo/+}$Ep-CAM$^+$ infundibulum and isthmus (IF & IS), Sca-1$^{lo}$Ep-CAM$^{lo}$ double negative (DN)) or CD45$^-$ dermal populations (CD31$^-$CD90$^+$CD140α$^+$ fibroblast, gp38$^+$CD31$^+$ lymphatic endothelial cells (LEC), gp38$^{lo}$CD31$^+$ blood endothelial cells (BEC), CD31$^-$CD90$^-$CD140α$^-$ double negative (DN)) from naïve WT mice (pooled from 5 mice/experiment). ND, not detected. See also Supplementary Fig. 5. Mean ± s.e.m. **(a,d)** Pooled from three experiments, **(c)** representative of two similar experiments. **(a)** One-way ANOVA with Bonferroni's multiple comparisons test, **(b,c)** unpaired two-tailed Student's *t*-test. *$P < 0.05$, **$P < 0.01$, ***$P < 0.001$, ****$P < 0.0001$.

ability to migrate toward CCL20 (Fig. 6b,c). Genetically marked control- and $Ccr6^{tg}$-transduced γδT17 cells were mixed 50:50 and transferred into B16 melanoma-bearing recipients. While the input ratio of transferred GFP$^-$ cells was maintained in all examined organs as expected, among GFP$^+$ cells, $Ccr6^{tg}$ γδT17

cells were enriched in the dermis but deficient in tumours (Fig. 6d). Similar results were observed during *S. pneumoniae* infection: $Ccr6^{tg}$ γδT17 cells were selectively deficient at homing to NP, but accumulated to a greater extent than control-transduced cells in uninflamed dermis (Fig. 6e). $Ccr6^{tg}$ γδT17

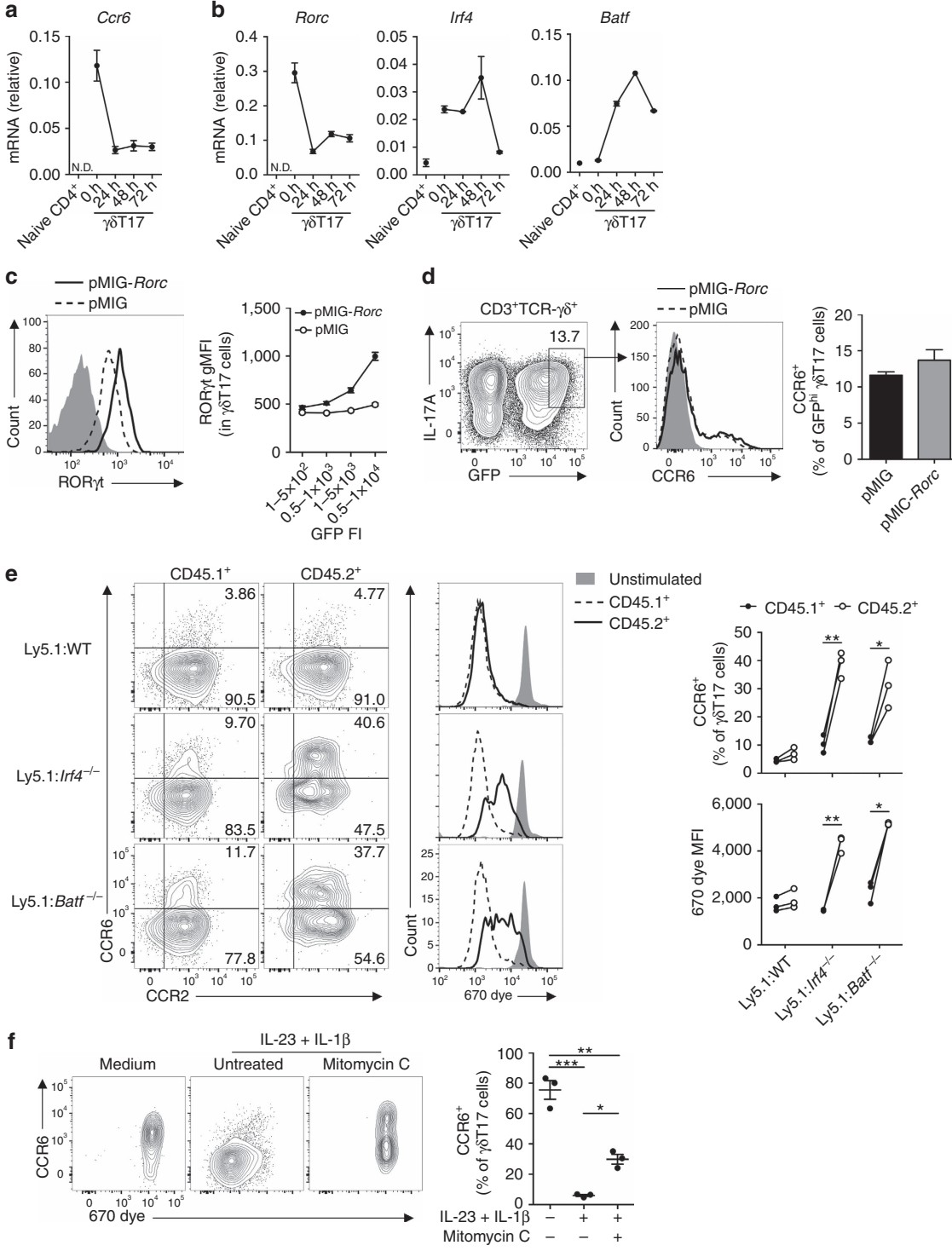

**Figure 5 | IRF4 and BATF promote CCR6 downregulation in γδT17 cells.** (**a**) *Ccr6* and (**b**) transcription factor mRNA in sorted γδT17 cells from *Il17a^Cre^ × Rosa26^eYFP^* lymphocytes *ex vivo* or cultured with IL-23/IL-1β for indicated times (pooled from 5 to 7 mice). ND, not detected. (**c,d**) Expanded γδT17 cells (*n* = 3) were transduced with empty pMIG or pMIG-*Rorc* retrovirus. (**c**) Representative flow cytometry of RORγt expression in GFP^hi^ γδT17 cells (gated as in **d**), relative to isotype (grey) and geometric mean fluorescence intensity (gMFI) relative to GFP fluorescence intensity (FI). (**d**) Representative flow cytometry of CCR6 expression and quantitation in GFP^hi^ γδT17 cells. (**e**) Splenocytes from Ly5.1 and either wild type (WT), *Irf4^−/−^* or *Batf^−/−^* mice were 670 dye-labelled, mixed 50:50 and stimulated with IL-23/IL-1β for 72 h. Representative flow cytometry and quantitation of CCR6 expression and proliferation in CD45.1^+^ or CD45.2^+^ γδT17 cells (*n* = 3/group). (**f**) Representative flow cytometry and quantitation of CCR6 expression by 670 dye-labelled γδT17 cells from WT splenocytes cultured with IL-23/IL-1β for 72 h with/without mitomycin C pre-treatment (*n* = 3). See also Supplementary Figs 6 and 7. (**a,b**) Mean ± s.d., (**c–f**) Mean ± s.e.m. (**a–f**) Representative of two similar experiments. (**d,e**) Paired two-tailed Student's *t*-test, (**f**) one-way paired ANOVA with Bonferroni's multiple comparisons test. *$P < 0.05$, **$P < 0.01$, ***$P < 0.001$.

cells also homed less efficiently to the CNS during EAE, although subcutaneous complete Freund's adjuvant immunization precluded analysis of homing to uninflamed skin in this model (Fig. 6f). Together, these experiments demonstrated that activated γδT17 cells with forced CCR6 expression were recruited to uninflamed dermis at the expense of homing to inflamed tissue.

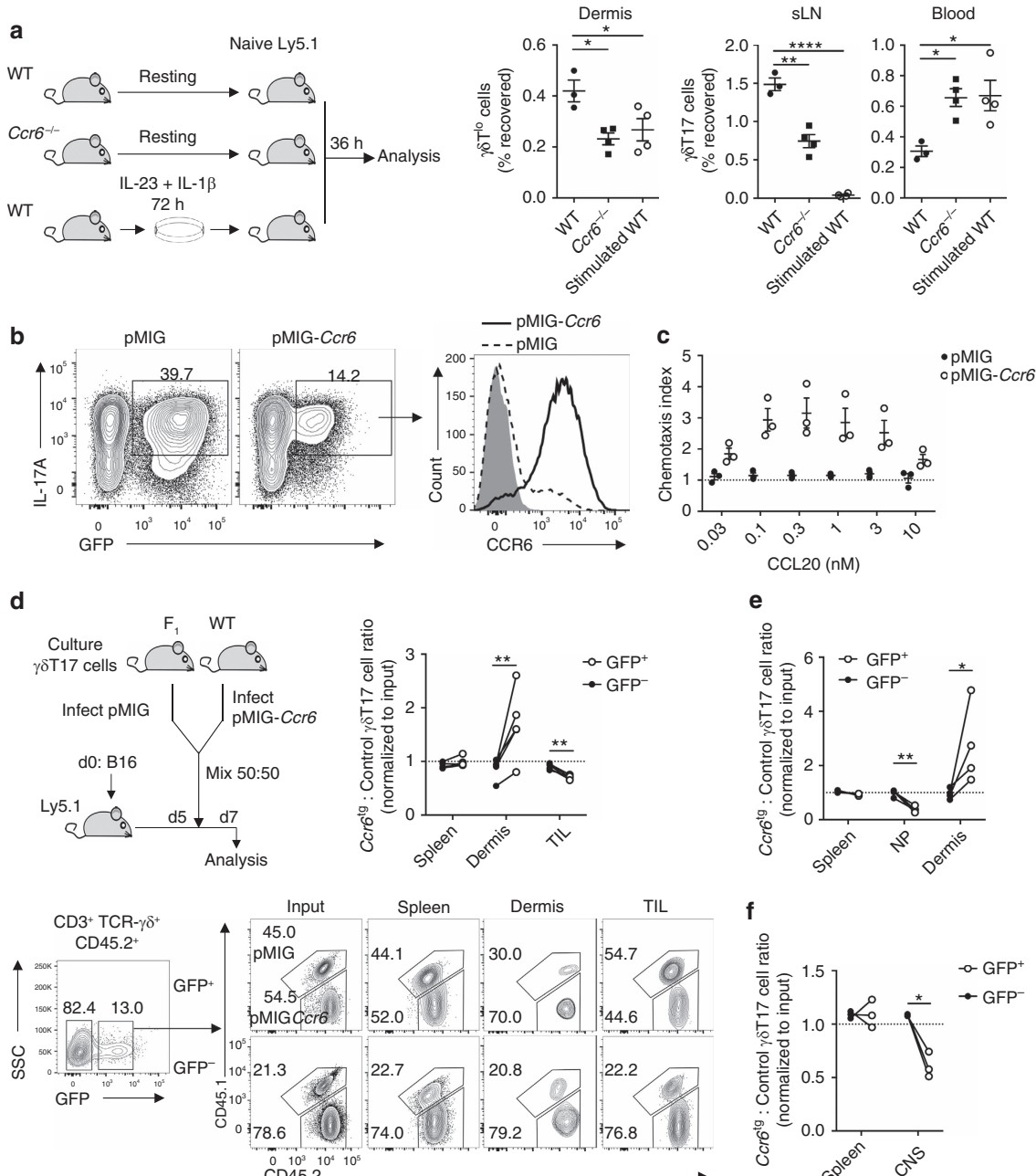

**Figure 6 | CCR6 downregulation by γδT17 cells enhances migration to inflamed tissue.** (**a**) Resting lymphocytes from wild type (WT) ($n = 3$) or $Ccr6^{-/-}$ ($n = 4$) mice, or WT lymphocytes stimulated with IL-23/IL-1β for 72 h ($n = 4$) were transferred i.v. into separate naïve Ly5.1 hosts. After 36 h, number of CD45.2$^+$ γδT$^{lo}$/γδT17 cells recovered was expressed as % of number transferred. sLN, skin-draining lymph node. (**b**) Representative flow cytometry for CCR6 expression by GFP$^+$ in vitro-expanded γδT17 cells transduced with empty pMIG or pMIG-Ccr6, relative to isotype (grey) ($n = 3$). (**c**) Chemotaxis of GFP$^+$ γδT17 cells transduced as in (**b**) to CCL20 ($n = 3$). (**d**) In vitro-expanded γδT17 cells from F$_1$ (CD45.1$^+$CD45.2$^+$) or WT (CD45.2$^+$) mice were transduced with empty pMIG or pMIG-Ccr6, respectively. Equal numbers of mixed GFP$^+$ cells were transferred i.v. into Ly5.1 mice challenged with B16 melanoma 5 days prior and analysed at d7 ($n = 5$). Representative flow cytometry and ratio of recovered F$_1$ to WT γδT17 cells within transduced (GFP$^+$) and untransduced (GFP$^-$) populations. Recovered values were normalized to input values. TIL, tumour-infiltrating lymphocytes. (**e,f**) In vitro-expanded γδT17 cells from WT or F$_1$ mice were transduced with empty pMIG or pMIG-Ccr6, respectively. Equal numbers of mixed GFP$^+$ cells were transferred i.v. into Ly5.1 mice either (**e**) 24 h post-infection with S. pneumoniae ($n = 4$) or (**f**) at experimental autoimmune encephalomyelitis (EAE) onset ($n = 3$) and organs were analysed 48 h later. Ratio of recovered WT to F$_1$ γδT17 cells within transduced (GFP$^+$) and untransduced (GFP$^-$) populations, normalized to input values. CNS, central nervous system; NP, nasal passage. Mean ± s.e.m. (**a**) Representative of three similar experiments, (**b,d**) representative of two experiments. (**a**) One-way ANOVA with Dunnett's multiple comparisons test relative to resting WT γδT17 cells, (**d–f**) paired two-tailed Student's t-test. *$P < 0.05$, **$P < 0.01$, ***$P < 0.001$, ****$P < 0.0001$.

Thus, CCR6 downregulation promotes γδT17 cell migration to inflammatory sites.

## Discussion

In the present study we define the molecular regulation of γδT17 cell trafficking between resting and activated states. Our data are consistent with a model in which γδT17 cells are imprinted with expression of both CCR6 and CCR2 during thymic development. CCR6 coordinates steady-state recruitment of circulating γδT17 cells into the dermis, where CCL20 is constitutively expressed, thus orchestrating their homeostatic recirculation. Upon activation, γδT17 cells rapidly downregulate CCR6 in an IRF4- and BATF-dependent manner. CCR2 expression is maintained and drives homing of activated γδT17 cells to inflamed tissue during autoimmunity, cancer and infection. CCR6 downregulation is required to promote optimal recruitment of activated γδT17 cells to inflamed tissue, by preventing their diversion and sequestration into uninflamed skin. These data identify a novel mode of lymphocyte trafficking that facilitates both γδT17 cell innate-like surveillance of host tissues and their rapid recruitment to distal sites of ongoing inflammation.

Our data demonstrate that CCR6 coordinates homeostatic recirculation of Vγ4[+] and Vγ6[+] γδT17 cells through blood, skin and sLNs. Given that γδT17 cells likely populate sLNs via afferent lymph[12,14], our data suggest that entry is contingent on first trafficking through the dermis, as Ccr6-deficient γδT17 cells fail to localize to both organs. Thus, in addition to driving infiltration of skin, CCR6 ultimately regulates γδT17 cell homing to sLNs. These findings complement a recent report that CCR6 positions sLN Vγ4[+] γδT17 cells in the subcapsular sinus[11]. Our findings are also in agreement with previous reports of steady-state γδT17 cell circulation, although we show that in addition to Vγ4[+], Vγ6[+] γδT17 cells also undergo constitutive trafficking[10,13,14]. While most Vγ4[+] γδT17 cells in sLNs do not recirculate within a 2-week period, approximately 25% undergo extensive circulation[11]. The non-recirculating population of sLN γδT17 cells is likely those positioned in interfollicular zones and subcapsular sinus, which were proposed to scan for sLN-invading microbes[11,46]. However, the function of the recirculating population of γδT17 cells is unclear. As γδT17 cells do not need to scan sLNs for antigen like conventional T cells, why they adopt a constitutive CCR6-dependent circulation loop between tissues and sLNs remains to be resolved.

We found that programmed expression of CCR2 equips γδT17 cells with the ability to rapidly home from circulation into diverse inflammatory environments. Our description of CCR2-driven γδT17 cell infiltration of the autoimmune CNS reflects other reports during psoriasis and arthritis[15,20]. Additionally, CCR2-mediated infiltration of both Vγ4[+] and Vγ6[+] γδT17 cells into B16 melanomas suggests that this axis is a universal inflammatory homing signal for γδT17 cells. Our data reveal a redundant function for CCR6 in these processes, due to its immediate downregulation upon γδT17 cell activation. However, these data are inconsistent with reports of CCR6-driven migration of γδT17 cells during skin and liver inflammation[19,47]. Tissue-specific signals directing maintained CCR6 expression in activated γδT17 cells in these particular scenarios could possibly explain discrepancies between these reports and our data, although this requires further investigation. The function of tissue-infiltrating γδT17 cells during inflammation in barrier tissues, where a resident γδT17 cell population already exists, is unclear. This phenomenon was previously reported during psoriasis, although did not appear to affect disease[14,48]. Here, we show that γδT17 cells expand in draining LNs and infiltrate nasal mucosa via CCR2 during bacterial infection. In absence of endogenous γδ T

cells, transferred circulating γδT17 cells are able to control infection in a CCR2-dependent manner. Thus, we propose that LN-expanded γδT17 cells home to lesions via CCR2 to supplement the local γδT17 cell pool during ongoing tissue inflammation.

Along with our previous work, this study identifies a shared chemokine receptor program used by IL-17-producing cells. We show that like Th17 cells[18], γδT17 cells express CCR6 early during their effector program, but lose CCR6 expression via IL-23 signalling during ongoing inflammation. Instead, CCR2 is the defining IL-17-program inflammatory homing signal. Whether CCR6 repression in Th17 cells occurs to disrupt barrier tissue-homing and promote recruitment to inflamed tissue, as we show in γδT17 cells, is unclear. Despite γδT17 cell development occurring independently of IRF4 and BATF[49,50], we report that these factors suppress Ccr6 expression in γδT17 cells upon IL-23- and IL-1β-driven activation. This is likely to be cooperative, as has been shown during Th17 cell development[43]. Moreover, we identified a common binding site for IRF4 and BATF in the Ccr6 promoter, suggesting that repression of Ccr6 expression is directly mediated by these factors. It remains to be seen if a similar mechanism operates in Th17 cells. Therefore, while γδT17 and Th17 cells both lose CCR6 expression during inflammation, the mechanism and function of this process may differ between these populations.

Investigating the trafficking of human γδT17 cells is of clinical relevance as they are increasingly implicated in autoimmunity and cancer, and it will be important to determine whether the model we have established here for murine γδT17 cells applies to humans. Of interest, CCR6 is known to be expressed by both Vδ1[+] and Vδ2[+] γδT17 cells in humans, as well as by circulating Vδ2[+] cells with a skin-homing CLA[+] phenotype[51–53]. CCR2 expression has been identified in human γδ T cells, but to our knowledge this has not been examined specifically in IL-17[+] cells[54,55]. Whether human γδT17 cells also undergo activation-induced CCR6 downregulation to enhance inflammatory homing has yet to be determined. While the human dermal γδT17 cell compartment is relatively small compared to the murine system, the relative abundance of circulating skin-homing Vδ2[+] cells suggests that similar recirculation mechanisms may also operate in humans[8,52]. These issues await further experimental resolution.

Our description of γδT17 cell trafficking between homeostasis and inflammation presents a novel mode of lymphocyte migration. Conventional T cells downregulate the homeostatic recirculation signal CCR7 and induce expression of inflammatory chemokine receptors upon differentiation into effector subsets, a slow process. In contrast, we show that γδT17 cells constitutively co-express homeostatic and inflammatory homing receptors. The switch in γδT17 cell trafficking from homeostatic to inflammatory programs is solely driven by downregulation of the homeostatic receptor CCR6, rather than induction of additional inflammatory homing receptors. This model likely facilitates immediate homing to inflammatory sites in addition to homeostatic scanning behaviour, consistent with the 'activated-but-resting' phenotype of γδT17 cells. Use of CCR6 expression to distinguish resting and activated states may facilitate future investigation of γδT17 cell biology. We conclude that γδT17 cells exhibit a unique bi-phasic trafficking program driven by programmed changes in homing receptor expression to facilitate tissue sentinel responses and rapid homing to distal inflammatory sites.

## Methods

**Mice.** C57Bl/6 (WT) and Ly5.1 mice were purchased from Animal Resource Centre (WA, Australia) or bred at the University of Adelaide animal facility. $Il17a^{Cre} \times Rosa26^{eYFP}$, $Ccr6^{-/-}$, $Ccr2^{-/-}$, $Ccr2^{-/-}Ccr6^{-/-}$, $Tcrd^{-/-}$ and

C57Bl/6 × Ly5.1 ($F_1$) mice were bred at the University of Adelaide animal facility. $Irf4^{-/-}$, $Irf8^{-/-}$, $Batf^{-/-}$ and $Lck^{Cre}Prdm1^{fl/fl}$ mice were bred at the WEHI animal facility. Mice were age- and gender-matched and used at 6–14 weeks of age. Experiments were conducted with approval of the University of Adelaide Animal Ethics Committee.

**Disease models.** Mice were immunized for chronic EAE by subcutaneous injection of 100 μg of $MOG_{35-55}$ (GL Biochem) in phosphate-buffered saline (PBS) emulsified 1:1 in complete Freund's adjuvant, coupled with i.p. injection of 300 ng Pertussis toxin (Sapphire Bioscience) on days 0 and 2. Mice were analysed at clinical scores of 0.5 (onset) and 2–3 (peak) in wild type (WT) mice, where scoring criteria were: 0.5 tremor, 1 partially limp tail, 2 fully limp tail, 2.25 unable to right, 2.5 sprawled hindlimbs, 2.75 one hindlimb paralysed, 3 both hindlimbs paralysed, 3.5 one forelimb paralysed. B16.F10 melanoma cells (provided by Prof. Mark Smyth, QIMR Berghofer, mycoplasma free and verified by short tandem repeat) were cultured in RPMI 1640 containing 10% fetal calf serum (FCS) and $5 \times 10^4$ cells were injected subcutaneously into mice at four sites. *S. pneumoniae* EF3030 was grown to a $D_{600}$ of 0.18 in nutrient broth with 10% horse serum at 37 °C 5% $CO_2$ and stored at $-80$ °C. Stocks were defrosted and $1 \times 10^6$ colony-forming units were delivered intranasally. Bacterial load was determined by serial dilution of concentrated NW onto blood agar with 5 μg ml$^{-1}$ gentamicin (Sigma).

**Cell preparation.** Single-cell suspensions were prepared from lymphoid organs by pressing through 70 μm filters. Peripheral blood was collected into heparinized Vacutainer tubes (BD). Red blood cells were lysed as required. CNS from PBS-perfused mice was pressed through 70 μm filters and then separated over a 30/70% Percoll (GE) gradient at 500$g$. The following digestions were performed at 37 °C with 30 U ml$^{-1}$ DNase (Sigma). Epidermis and dermis from ears or shaved trunk skin were separated by incubation in 0.375% tryspin for 2 h at 37 °C, and then digested separately with 85 μg ml$^{-1}$ Liberase TM (Roche) for 1 h. Tumours and perfused lungs were digested in 2 mg ml$^{-1}$ collagenase (Sigma) for 1 h. NP tissue between the nose tip and eyes was dissected following removal of nasal-associated lymphoid tissue, digested with 2 mg ml$^{-1}$ collagenase for 1.5 h and separated by a 40/80% Percoll gradient at 600$g$. Livers were pressed through 70 μm filters and then lymphocytes were isolated by a 37.5% Percoll gradient at 850$g$. Flushed and longitudinally opened small intestine free of Peyer's patches was washed in PBS then incubated in 5 mM EDTA for 40 min at 37 °C to remove epithelium, before remaining lamina propria was digested with 0.5 mg ml$^{-1}$ collagenase for 1.5 h at 37 °C and separated over a 40/80% Percoll gradient at 1,000$g$ to isolate lamina propria lymphocytes. Peritoneal exudate cells were collected by $3 \times 1$ ml PBS washes.

**Flow cytometry.** Single-cell suspensions were stained in 96-well round- or v-bottom plates (Corning) at $2 \times 10^6$ lymphocytes per well using antibodies and other reagents detailed in Supplementary Table 1. For intracellular cytokine staining, cells were first incubated in complete IMDM with 20 pg ml$^{-1}$ PMA (Life Technologies), 1 nM ionomycin (Life Technologies) and 1/1,500 GolgiStop (BD) for 4 h at 37 °C. Cells were washed in PBS and stained with Near Infrared fixable dye diluted 1/1,000 (Life Technologies) for 15 min at room temperature. Cells were then washed with FACS buffer (PBS 1% bovine serum albumin 0.04% azide) and blocked with mouse γ-globulin (mγg) (200 μg ml$^{-1}$) for 5 min at room temperature. All subsequent steps were incubated at 4 °C. For purified antibodies, cells were stained with purified antibody for 20–60 min, washed in FACS buffer, stained with secondary antibody in mγg (200 μg ml$^{-1}$) and normal mouse serum (NMS) (1%) for 20 min, washed in FACS buffer and blocked with rat γ-globulin for 15 min. Cells were stained with directly conjugated and biotinylated antibodies for 20 min. In the case of biotinylated antibodies, cells were then washed in FACS buffer and stained with streptavidin for 15 min. Cells were then washed in PBS 0.04% azide. For intracellular cytokine staining, cells were incubated in Cytofix/Cytoperm (BD) for 20 min, washed in Permwash (BD) and stained with intracellular directly conjugated antibodies for 20 min. For transcription factor staining, cells were incubated in Foxp3 kit perm buffer (eBioscience) for 30 min to overnight, and then washed in Foxp3 kit permwash (eBioscience). Cells were then stained with directly conjugated α-transcription factor antibodies in NMS (2%) and normal rat serum (2%), and then washed in PBS 0.04% azide. All stains were washed in PBS 0.04% azide, resuspended in PBS 1% paraformaldehyde and stored at 4 °C in the dark.

Specificity of α-CCR2 was confirmed by negative staining on $Ccr2^{-/-}$ γδT17 cells. CCR2 and CCR6 gating was determined by relevant isotype controls. For measurement of fluorescence intensity, relevant isotype control geometric mean fluorescence intensity was subtracted from raw geometric mean fluorescence intensity value. For *in vivo* proliferation analysis, mice were given 2 mg BrdU i.p. and then drinking water with 0.8 mg ml$^{-1}$ BrdU 2% glucose. Following restimulation and surface staining, cells were permeabilized, DNase-treated and stained with α-BrdU (BD) according to the manufacturer's instructions. For *in vitro* proliferation analysis, cells were labelled with Cell Proliferation Dye (eBioscience) according to the manufacturer's instructions. Flow cytometry was

acquired on a BD LSR II or FACSAria and analysed with FlowJo (Treestar). Gating strategies are detailed in Supplementary Fig. 8.

**γδT17 cell expansion culture and retroviral transduction.** Pooled spleen and LN cells were cultured at $1 \times 10^6$ cells per ml in RPMI 1640 containing 10% FCS, antibiotics, 1× Glutamax (Gibco), 10 mM HEPES (SA Pathology), 1 mM sodium pyruvate, 54 pM β-mercaptoethanol and 1× non-essential amino acids (Gibco) with 5 ng ml$^{-1}$ recombinant (r)IL-23 (eBioscience), 5 ng ml$^{-1}$ rIL-1β (Miltenyi Biotec) and 10 μg ml$^{-1}$ α-IFN-γ (BioXCell) in 96-well round-bottom plates coated with 1 μg ml$^{-1}$ α-TCR-γδ (clone GL3; Biolegend) for 3 days. Cells were washed and re-seeded on fresh plastic at $1 \times 10^6$ cells per ml for a further 3 days as above without TCR-γδ stimulation. Cells were then washed and re-seeded in 20 ng ml$^{-1}$ rIL-7 (Peprotech) and 10 μg ml$^{-1}$ α-IFN-γ for a further 3 days. pMIG, pMIG-*Rorc* and pMIG-*Ccr6* (cloned from mouse *Ccr6* cDNA) were transfected into EcoPack 2 293 cells (Clontech; mycoplasma free) with Lipofectamine 2000 (ThermoFisher), and supernatant collected after 48 h. γδT17 cells at days 4 and 5 of culture were centrifuged at 2,500 r.p.m. (30 °C for 1.5 h) in supernatant with 8 μg ml$^{-1}$ polybrene (Sigma) in flat-bottom 96 well trays before being returned to culture.

**Adoptive transfers.** For $Ccr2^{-/-}$ and $Ccr6^{tg}$ trafficking experiments, $1–2 \times 10^6$ each of *in vitro*-expanded $F_1$ (CD45.1$^+$CD45.2$^+$) and $Ccr2^{-/-}$ (CD45.2$^+$), or transduced control and $Ccr6^{tg}$ γδT17 cells ($F_1$ or WT), were mixed and transferred i.v. into Ly5.1 (CD45.1$^+$) recipient mice d5 post-challenge with B16 melanoma, d8–10 post EAE induction or 24 h post-*S. pneumoniae* infection. γδT17 cell infiltration of target organs was analysed 24–48 h post-transfer, and CD45 congenic ratios were normalized to input sample. For *S. pneumoniae* $Tcrd^{-/-}$ reconstitution, *in vitro*-expanded WT and $Ccr2^{-/-}$ γδT17 cells were further purified by MACS (Miltenyi Biotec) before $3 \times 10^6$ cells were transferred into separate $Tcrd^{-/-}$ hosts 24 h prior to infection. For naïve dermis trafficking experiments, $5–10 \times 10^7$ fresh WT or $Ccr6^{-/-}$ lymphocytes or $3 \times 10^7$ 72 h IL-23/IL-1β-stimulated WT lymphocytes were transferred into separate unimmunized Ly5.1 mice and analysed 36 h later. Number of recovered cells was normalized to number of γδT17 cells transferred.

**ELISA.** Tumour and NP supernatants from digested samples and supernatants from filtered CNS were supplemented with protease inhibitors (Sigma) and stored at $-80$ °C. Mouse CCL2 Duoset ELISA (R&D) was conducted according to the manufacturer's instructions.

**qPCR.** γδ T cells from $Il17a^{Cre} \times Rose26^{eYFP}$ mice were enriched by MACS using mouse TCRγδ$^+$ isolation kit (Miltenyi Biotec, # 130-092-125), and then sorted using a BD FACSAria. Naïve CD4$^+$ T cells were sorted from WT splenocytes, and skin stromal populations were sorted from digested epidermal and dermal suspensions from WT mice. Sorting strategies are detailed in Supplementary Figure 9. RNA was extracted from sorted cells using Qiagen RNeasy Micro kit (# 74004). For epidermis and dermis, tissues were snap frozen in liquid nitrogen, crushed with mortar and pestle and RNA purified using Qiagen RNeasy Mini kit (# 74104) according to the manufacturer's instructions. cDNA was generated with the Roche Transcriptor First Strand cDNA synthesis kit (# 04896866001). qPCR was performed with Roche LightCycler 480 SYBR Green I master mix (# 04887352001) using primer sequences in Supplementary Table 2 on a LightCycler 480 instrument (Roche). Relative gene expression was calculated as $2^{-(CT\ target–CT\ reference)}$ where reference was *Rplp0*.

**Chemotaxis.** Splenocytes were rested in complete RPMI for 3–4 h at 37 °C, washed and suspended in chemotaxis buffer (RPMI 0.5% bovine serum albumin 20 mM HEPES). γδT17 cells from culture were washed and suspended in chemotaxis buffer. CCL2 or CCL20 (from the late Prof. Ian Clark-Lewis) were diluted in chemotaxis buffer and loaded into the lower chambers of 96-well 5 μm pore transwell plates (Corning). $2 \times 10^6$ splenocytes or $2 \times 10^5$ *in vitro*-expanded γδT17 cells were loaded into the upper chambers and plates were incubated at 37 °C for 3 h. Lower wells were harvested and stained for flow cytometry. CountBrite beads (Invitrogen) were added to samples prior to acquisition to normalize event counts. Chemotaxis index was calculated as number of gated events divided by number in 0 chemokine control.

***In vitro* stimulation.** Splenocytes were cultured in complete IMDM (10% FCS, pen/strep, L-glutamine, β-mercaptoethanol) at $2.5 \times 10^6$ cells per ml with 10 ng ml$^{-1}$ rIL-23 (eBioscience), 10 ng ml$^{-1}$ rIL-1β (Miltenyi Biotec), 20 ng ml$^{-1}$ rIL-7 (Peprotech), 10 ng ml$^{-1}$ rIL-12 (R&D), 10 ng ml$^{-1}$ rIL-18 (R&D) and/or with pre-coating in 1 μg ml$^{-1}$ α-TCR-γδ (Biolegend) for up to 72 h at 37 °C. For mitomycin C pre-treatment, cells were first incubated with 10 μg ml$^{-1}$ mitomycin C (Sigma) in complete IMDM at $2 \times 10^7$ cells per ml for 2 h at 37 °C before extensive washing.

**ChIP-seq analysis.** ChIP-seq data for IRF4 in CD8$^+$ T cells[45], BATF in CD8$^+$ T cells[44] and IRF4/BATF in Th17 cells[43] were previously published and were

obtained from NCBI database using accession codes GSE49930, GSE54191 and GSE40918, respectively. BAM files were loaded and displayed using the IGB genome browser.

**Statistics.** Data were analysed with GraphPad Prism 6. Appropriate statistical tests were two-sided and used as indicated in figure legends. $*P < 0.05$, $**P < 0.01$, $***P < 0.001$, $****P < 0.0001$. All replicates are biological except in qPCR experiments, where technical replicates are denoted. Sample sizes were determined empirically to ensure adequate power. $Tcrd^{-/-}$ host mice were randomly assigned to groups before receiving adoptive transfers in Fig. 3f. No blinding was utilized. Minimal variance was generally observed between groups; Welch's correction was used in $t$-tests where standard deviations were significantly different. Most data sets were normally distributed; Mann–Whitney and Kruskal–Wallis tests were utilized where data were not normally distributed.

**Data availability.** Relevant data are available from authors upon reasonable request.

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

## Acknowledgements

We thank Prof. Ian Frazer (UQ Diamantina Institute, Australia), Prof. Brigitta Stockinger (Francis Crick Institute, UK) and Prof. Christian Engwerda (QIMR Berghofer, Australia) for mice and Prof. Dan Littman (New York University, USA) for pMIG-*Rorc* (Addgene #24069). We also thank the late Prof. Ian Clark-Lewis for chemokine ligands and staff at Laboratory Animal Services from University of Adelaide for animal husbandry. This work was supported by National Health and Medical Research Council project grants 1066781 and 1054925. A.K. is supported by the Sylvia and Charles Viertel foundation.

## Author contributions

D.R.M., E.E.K., I.C. and S.R.M. designed experiments, analysed data and wrote the manuscript; D.R.M. performed most experiments; E.E.K., C.R.B., T.S.T., K.A.F., C.E.G. and J.J.W. performed experiments; R.B. and J.C.P. contributed to *S. pneumoniae* experiments; A.K. and S.L.N. contributed to transcription factor experiments; A.B. contributed to retrovirus experiments; M.M. provided key reagents; J.C.P., A.K., S.L.N., A.B. and M.M. edited the manuscript; I.C. and S.R.M. supervised the study and attained funding.

## Additional information

**Competing interests:** The authors declare no competing financial interests.

