## [Peer Review File · Nature Communications]

Reviewers' comments:

Reviewer #1 (Remarks to the Author):gd, EAE, infectious and tumor model

It is hard to find the single big new finding in this paper. The experiments described show that CNS infiltrating $\gamma\delta$ T17 cells during EAE were largely CCR6-CCR2+ cells and that these cells appear to be the central in bacterial clearance and immunity to tumours. However it has previously been reported that CCR6 and NK1.1 distinguish IL-17 and IFN- γ producing $\gamma\delta$ T cells (Haas et al, Immunity 2012). Furthermore it has been reported that 'CCR6 is required for epidermal trafficking of $\gamma\delta$ T cells in an IL-23-induced model of psoriasisiform dermatitis' (Mabuchi et al J. Invest. Dermatol, 2013). The authors of the present study use animal models of autoimmunity, infection and cancer and consequently the paper is very disjointed and difficult to see the logic of one experiment to the next. Furthermore there are a significant number of unanswered questions

Specific comments

1. What is the frequency of CCR6+CCR2- $\gamma\delta$ T cells in adult thymus, iLN and CNS in mice with EAE at onset and peak of disease? Was CCR6 and CCR2 expression assessed on $\alpha\beta$ T cells?
2. It would be good to examine the contribution of CCR6+ or CCR2+ $\gamma\delta$ T cells in the amplification of Th17 cells using the adoptive transfer model of EAE.
3. The data demonstrates that TCR stimulation downregulates CCR6 expression on $\gamma\delta$ T cells; does stimulation with IL-12/IL-18 upregulate CCR6 or CCR2 expression?
4. Markers of T cell activation, such as CD25, CD69, CD44 or CD38 should be assessed in conjunction with downregulation of CCR6 expression.
5. The data suggests that CCR6 regulates the retention of dermal $\gamma\delta$ T17 cells. It would be nice to assess if CCR2-/- $\gamma\delta$ T cells effective/defective in trafficking in models of psoriasis
6. Do $\gamma\delta$ T cell subsets have differential expression of CCR6 and CCR2 during disease?
7. The data in Figure 1e and figure 5f are conflicting. 'CCR6 expression was downregulated in proliferated $\gamma\delta$ T cells' 'CCR6 downregulation still occurred upon Mitomycin treatment, suggesting that proliferation and CCR6 downregulation were coincident but independent". The authors should include mitomycin C treatment with BrdU incorporation to demonstrate that downregulation of CCR6 in $\gamma\delta$ T cells is independent of proliferation and only through activation.
8. The data in the melanoma and EAE models suggest that CCR6 is not required for cellular trafficking, however CCR2 is essential. In the S. pneumonia infection model, it is unclear if CCR6 expression is required for protective $\gamma\delta$ T cell responses as only the effects with CCR2-/- $\gamma\delta$ T cell transfer were assessed.

Reviewer #2 (Remarks to the Author):gd T

This study by McKenzie et al defines a differential role for CCR2 and CCR6 in gdT17 homing to inflamed tissues and the dermis, respectively. Upon activation, the gdT17 cells downregulate CCR6, releasing them to migrate to inflamed sites via CCR2. I found this study most interesting and it clearly represents a robust body of work. The model proposed by the authors may appeal to a broader audience particularly since this type of "release" mechanism appears quite novel.

I have some minor comments on this study:

1. Fig 2 - Is there an effect of CCR6, CCR2 and CCR2.CCR6 KO on tumor growth/EAE scores?
2. Fig 2 vs Fig.4. It is not clear why in some cases the authors use CD3-bright gdT17 cells whereas

others CD3- $\text{I}\alpha$.

3. Whilst the importance of human gdT17 cells is less clear, it is an obvious question that may be worth discussing. Eg do any human gdT cells express these chemokine receptors and are they modulated by stimulation?

Reviewer #3 (Remarks to the Author):gd17, cytokine

Authors Duncan R.McKenzie et al. describe "IL-17-producing $\gamma\delta$ T cells switch migratory pattern between resting and activated states". Authors found that "naturally occurring" IL-17A-eYFP+ $\gamma\delta$ T cells coexpress CCR2 and CCR6 in thymus, spleen and LN, irrespective of V γ 4 or V γ 6 expression, and the CCR6-CCR2+ IL-17A-eYFP+ $\gamma\delta$ T cells were increased in non-lymphoid organs such as , lung , PP and si LPL of naive mice and in the LN and CNS after induction of inflammation such as EAE, Melanoma and S.pneumoniae infection. CCR6 are down regulated in Brdu+ IL-17A-eYFP+ $\gamma\delta$ T cells during inflammation or after in vitro stimulation, presumably via BATF and IRF4 induction. This is well-performed paper and their findings are interesting and informative for localization of IL-17-producing $\gamma\delta$ T cells in inflamed tissues. My comments are as follows:

General comment.

Notable finding in this paper are in vivo significance of down regulation of CCR6 activated $\gamma\delta$ T cells and its molecular mechanism. However, authors show only difference in Fig 6, in which CCR6 transduced IL-17A-eYFP+ $\gamma\delta$ T cells migrated away from tumor. Authors also should do this experiment in EAE and S. pneumoniae infection. For molecular mechanism, author show only descriptive data that BATFKO or IRF-4 KO IL-17A-eYFP+ $\gamma\delta$ T cells fail to downregulate CCR6 only partly after in vitro stimulation with IL-23 and IL1beta.

Specific comments

#1. Figure 2: Score of EAE may be ameliorated in CCR2-/- mice. Tumor size may be enlarged in CCR2-/- mice. Authors show the progress in diseases in CCR2-/- mice There are only small difference in migration in inflamed sites between activated CCR2-/- and CCR2+/+ IL-17A-eYFP+ $\gamma\delta$ T cells. FACS profile in EAE should be shown in transfer experiments.

#2. Figure 5. Author show only descriptive data that BATFKO or IRF-4 KO IL-17A-eYFP+ $\gamma\delta$ T cells fail to down-regulate CCR6 after in vitro stimulation with IL-23 and IL1b. BATF is basic leucine zipper protein that dimerizes with JUN proteins to the transcriptional activity of functionally important genes in lymphocytes for functional maturation and migration of effector T cells. How BATF or IRF4 down-regulate CCR6? Author conclude that down-regulation of CCR6 occurs independent to cell proliferation. Dose Mytomycin C inhibit cell proliferation sufficiently? Do BATFKO or IRF-4KO $\gamma\delta$ T cells express IL23R, IL1R, or IL-17A normally? BATF regulates localization of $\alpha\beta$ T cells in intestinal niches via directing expression of CD103 and CCR9 (J.E.M. 210:475-489, 2010). BALB KO mice may show an abnormal localization of $\gamma\delta$ T cells. CCR6 promoter/enhancer assay may clarify the direct effect of these molecules on cytokines-induced down-regulation.

#3 Figure 6: This is one of the most important figures in this paper, showing in vivo a significance of down regulation of CCR6 on $\gamma\delta$ T cells for preventing sequestration into uninflamed dermis from inflamed tissues. However, authors show only minor difference in Fig 6, in which CCR6-transduced IL-

17A-eYFP+ $\gamma\delta$ T cells migrated away from tumor. Author should show that the CCR6-transduced IL-17A-eYFP+ $\gamma\delta$ T cells $\gamma\delta$ T cell in the dermis Authors also should do this experiment in EAE and S. pneumoniae infection and the $\gamma\delta$ T cell in the dermis

Minor comments

#1 Introduction: many references are misscited. Authors should cite more original papers.

#2 Figure 4: What is the role of CCR6 on IL-17A-eYFP+ $\gamma\delta$ T cells except migration to dermis? Is there any difference in localization of CCR6-/- $\gamma\delta$ T cells in other organs.

REVIEWERS' COMMENTS:

Reviewer #1 (Remarks to the Author):

The authors have addressed all my comments either by arguing them or with additional data. While I don't accept all the arguments, the manuscript is improved and represents a nice body of work.

Reviewer #2 (Remarks to the Author):

I thank the authors for their rebuttal. I am satisfied with this response and have no further comments.

Reviewer #3 (Remarks to the Author):

Authors responded satisfactorily to the reviewer's comments and suggestions.

Reviewers' comments:

Reviewer #1 (Remarks to the Author):gd, EAE, infectious and tumor model

It is hard to find the single big new finding in this paper.

The major finding is outlined in the title: $\gamma\delta$ T17 cells switch their trafficking pattern between resting and activated states. This is entirely new knowledge and establishes a new paradigm in T cell trafficking, whereby $\gamma\delta$ T17 co-express inflammatory (CCR2) and homeostatic (CCR6) chemokine receptors and respond to inflammatory stimulus by losing the homeostatic migratory receptor enabling them to be rapidly recruited to inflamed sites. In terms of T cell homing, this is a novel mode of control over migration, which we are convinced will be of significant interest to the field. To quote the two other expert reviewers who critiqued this study: “this study is most interesting and it clearly represents a robust body of work. The model proposed by the authors may appeal to a broader audience particularly since this type of “release” mechanism appears quite novel” (reviewer 2); and “This is well-performed paper and their findings are interesting and informative for localization of IL-17-producing $\gamma\delta$ T cells in inflamed tissues” (reviewer 3).

To reiterate, this broad finding encompasses the following specific novel findings:

- $\gamma\delta$ T17 cells switch off CCR6 expression during activation (via IRF4/BATF signaling)
- CCR2, but not CCR6, is the universal inflammatory homing receptor for $\gamma\delta$ T17 cells
- downregulation of CCR6 enhances $\gamma\delta$ T17 cell homing to inflamed tissue by disrupting their homeostatic circuit through uninflamed skin

The experiments described show that CNS infiltrating $\gamma\delta$ T17 cells during EAE were largely CCR6-CCR2+ cells and that these cells appear to be the central in bacterial clearance and immunity to tumours. However it has previously been reported that CCR6 and NK1.1 distinguish IL-17 and IFN- γ producing $\gamma\delta$ T cells (Haas et al, Immunity 2012). Furthermore it has been reported that ‘CCR6 is required for epidermal trafficking of $\gamma\delta$ T cells in an IL-23-induced model of psoriasiform dermatitis’ (Mabuchi et al J. Invest. Dermatol, 2013).

The reviewer seems to be suggesting a lack of novelty in our findings with these remarks, but may be largely missing the major point of our study. We make no claim to be the first to describe CCR6 as a marker of $\gamma\delta$ T17 cells or to be the first to show that CCR6 regulates $\gamma\delta$ T17 cell trafficking in some way. Our data substantially build upon the report by Haas *et al.* to provide entirely novel information that changes the concept of how $\gamma\delta$ T17 cells regulate their migration. We show that $\gamma\delta$ T17 cells express CCR2 along with CCR6 at rest, but during activation (either *in vitro* or *in vivo*) we report that CCR6 expression is lost (Fig 1d-f). This highlights that the CCR6-based marker strategy of Haas *et al.* is only relevant in unimmunized mice; our description of CCR2 expression, which is maintained between resting and activated states, indicates that CCR2 is a superior surface marker of $\gamma\delta$ T17 cells. Our findings regarding the role of CCR6 in $\gamma\delta$ T17 cell trafficking to inflamed tissue are somewhat different to conclusions drawn by Mabuchi *et al.*, who reported that dermal $\gamma\delta$ T cells use CCR6 to enter the inflamed epidermis during skin inflammation. Our data support a role for CCR6 in homeostatic recruitment of $\gamma\delta$ T17 cells to the uninflamed skin but

clearly show that the activity of CCR6 is lost in activated settings. It is important to note that Mabuchi *et al.* investigated mice globally deficient in CCR6 signaling (*Ccr6*^{-/-} or CCL20 neutralization) to form their conclusions, while we have employed cell-intrinsic systems, which enables more accurate investigation of $\gamma\delta$ T17 cell trafficking. Additionally, we utilized distinct inflammatory scenarios from Mabuchi *et al.* (see point 5), which may further explain these differences due to possible tissue- and model-specific factors. The submitted manuscript highlighted the discrepancies between the Mabuchi report and our data, but we have now incorporated more discussion of this point (lines 341-345).

The authors of the present study use animal models of autoimmunity, infection and cancer and consequently the paper is very disjointed and difficult to see the logic of one experiment to the next.

We believe that the use of animal models of autoimmunity, cancer and infection to investigate the broad concepts of $\gamma\delta$ T17 cell trafficking is a strength of the paper, as it allows us to extending our findings from model-specific phenomena to broad underlying principles of $\gamma\delta$ T17 cell biology. We consider that the selection of these models is logical:

- EAE is initially used to examine activation of $\gamma\delta$ T17 cells, as it is a well-characterised model in this regard, and extends our recently published findings with Th17 cells in this model (Kara *et al.* 2015)
- we then use EAE and cancer to study $\gamma\delta$ T17 cell trafficking in scenarios where local tissue-resident $\gamma\delta$ T17 cells are absent, so that the analysis focuses on migrated cells
- subsequently, we extend the finding that CCR2 recruits $\gamma\delta$ T17 cells to these scenarios to an infectious setting that involves a local resident population, examining $\gamma\delta$ T17 trafficking-dependent protection in *S. pneumoniae* infection
- examining $\gamma\delta$ T17 cell trafficking into uninflamed skin enables us to study their homeostatic trafficking patterns as the skin is a major homeostatic reservoir of these cells
- finally, we return to inflammation to study why CCR6 expression is lost upon $\gamma\delta$ T17 cell activation – we examined trafficking of *Ccr6*^{tg} $\gamma\delta$ T17 cells during cancer in the submitted manuscript, but as requested by reviewer 3, have extended these experiments to include EAE and *S. pneumoniae* upon revision to more broadly apply our findings (Fig 6e-f)

We have modified the results section to more clearly link one experiment to the next (lines 162-163, 220-222).

Furthermore there are a significant number of unanswered questions
Specific comments

1. What is the frequency of CCR6+CCR2- $\gamma\delta$ T cells in adult thymus, iLN and CNS in mice with EAE at onset and peak of disease? Was CCR6 and CCR2 expression assessed on $\alpha\beta$ T cells?

As is evident from Fig 1a, c-d, f in the submitted manuscript, CCR2⁻ $\gamma\delta$ T17 cells are negligible in thymus, iLN, CNS and all tissues, and this does not change during EAE. We have modified the text to emphasize this observation more strongly (lines 119-

120). CCR6 and CCR2 in $\alpha\beta$ T cells was entire the focus of our recent paper (Kara *et al.* 2015, Nat Commun, ref #18), which provided the basis for the present study as was clearly written in the manuscript (lines 65-69, 81-85).

2. It would be good to examine the contribution of CCR6+ or CCR2+ $\gamma\delta$ T cells in the amplification of Th17 cells using the adoptive transfer model of EAE.

Please note that we are not studying the migration of $\gamma\delta$ T cells generally. Rather, we are studying the IL-17-producing subset ($\gamma\delta$ T17 cells). Only this subset expresses CCR6 and/or CCR2 (Fig 1b, Fig S2a), therefore $\gamma\delta$ T17 cells would be the only relevant cells for such an experiment. However, as is a key message of the paper, $\gamma\delta$ T17 cells are either CCR6⁺CCR2⁺ (resting) or CCR6⁻CCR2⁺ (activated). Therefore, it is not possible to separate CCR6⁺ from CCR2⁺ $\gamma\delta$ T17 cells as proposed by the reviewer, so the experiment is not possible.

The data demonstrates that TCR stimulation downregulates CCR6 expression on $\gamma\delta$ T cells; does stimulation with IL-12/IL-18 upregulate CCR6 or CCR2 expression?

We have conducted this experiment and thank the reviewer for the suggestion. We have included the data in the revised manuscript (Fig S2b – see below). IL-18 had negligible effects by itself, but synergized with IL-23 to promote CCR6 downregulation in $\gamma\delta$ T17 cells, consistent with its published role in activating these cells in combination with IL-23 (Lalor *et al.* 2011 J. Immunol). IL-12 did not affect CCR6 expression, consistent with the lack of IL-12R expression by $\gamma\delta$ T17 cells (Kulig *et al.* 2016 Nat Commun) and no effect of IL-12 upon activation of $\gamma\delta$ T17 cells (Petermann *et al.* 2010 Immunity). Therefore we maintain our conclusion that activation of $\gamma\delta$ T17 cells, via any known pathway, promotes repression of CCR6.

Figure S2 b) Representative flow cytometry of CCR6/CCR2 expression by $\gamma\delta$ T17 cells from *III17a^{Cre} x Rosa26^{eYFP}* lymphocytes cultured with indicated stimuli for 72 hr (n=3).

3. Markers of T cell activation, such as CD25, CD69, CD44 or CD38 should be assessed in conjunction with downregulation of CCR6 expression.

We have conducted this experiment. As is shown below and now included in the manuscript (Fig S2c), IL-23 and IL-1 β -stimulation drove upregulation of activation markers CD25, CD44 and CD69 in $\gamma\delta$ T17 cells, concomitantly with CCR6 downregulation. Marker CD38 did not change substantially upon stimulation (although its relevance to $\gamma\delta$ T17 cell biology is unknown). This result further supports our conclusion that activation drives repression of CCR6 expression in $\gamma\delta$ T17 cells.

Figure S2 c) Representative flow cytometry of activation marker expression by $\gamma\delta$ T17 cells from *Il17a^{Cre}xRosa26^{YFP}* lymphocytes either unstimulated or IL-23/IL-1 β stimulated for 72 hr (n=3).

- The data suggests that CCR6 regulates the retention of dermal $\gamma\delta$ T17 cells. It would be nice to assess if CCR2^{-/-} $\gamma\delta$ T cells effective/defective in trafficking in models of psoriasis

This has already been demonstrated in the literature by Jason Cyster's group (Ramírez-Valle *et al.* 2015). We have repeatedly referenced this finding in the submitted manuscript (reference #15, lines 72-75, 335-337). They demonstrated that CCR2 indeed was required for recruitment of LN-expanded $\gamma\delta$ T17 cells to the inflamed skin during experimental psoriasis.

- Do $\gamma\delta$ T cell subsets have differential expression of CCR6 and CCR2 during disease?

The only $\gamma\delta$ T cell subset to express CCR6 and/or CCR2 is the $\gamma\delta$ T17 subset. Figure 1b in the submitted manuscript demonstrated that non-IL-17-producing $\gamma\delta$ T cells do not migrate to either CCL20 or CCL2. We now include data below and in the manuscript (Fig S2a, lines 91-92, 108-110) demonstrating that while CD44^{hi}CD27⁻ $\gamma\delta$ T cells (i.e. $\gamma\delta$ T17 cells) are CCR6⁺CCR2⁺ at rest and become CCR6⁻CCR2⁺ during EAE, other $\gamma\delta$ T cell subsets (CD44^{hi}CD27⁺ and CD44^{lo}CD27⁺) are CCR6⁻CCR2⁻ at rest and remain so during EAE. Thus, CCR6/CCR2 expression is restricted to IL-17-producing $\gamma\delta$ T cells at homeostasis and during inflammation.

Figure S2 a) Representative flow cytometry of CCR6/CCR2 expression by subsets of $\gamma\delta$ T cells distinguished by CD44 and CD27 expression in sLNs of WT mice either naïve or at EAE onset (n=3/group).

- The data in Figure 1e and figure 5f are conflicting. 'CCR6 expression was downregulated in proliferated $\gamma\delta$ T cells' 'CCR6 downregulation still occurred upon Mitomycin treatment, suggesting that proliferation and CCR6 downregulation were coincident but independent''.

These data are not at all conflicting; rather, they are complementary. Fig 1e demonstrates that BrdU⁺ $\gamma\delta$ T17 cells during EAE are largely CCR6⁻, suggesting that proliferated (hence activated) $\gamma\delta$ T17 cells lose CCR6 expression. To then separate activation and proliferation (a product of activation), Figure 5f used Mitomycin C treatment to block proliferation in stimulated $\gamma\delta$ T17 cells. As CCR6 was downregulated upon stimulation even when proliferation was blocked, we conclude that activation induces both proliferation and loss of CCR6 in $\gamma\delta$ T17 cells (entirely consistent with Figure 1e), but that proliferation itself is not the driver of CCR6 downregulation, rather it is coincident with the phenomenon.

The authors should include mitomycin C treatment with BrdU incorporation to demonstrate that downregulation of CCR6 in $\gamma\delta$ T cells is independent of proliferation and only through activation.

The submitted manuscript used a cell proliferation dye 'eFluor 670' for this purpose in Figure 5f. As is evident from the flow cytometry plot, unstimulated $\gamma\delta$ T17 cells were CCR6⁺ and did not divide. Stimulated $\gamma\delta$ T17 cells proliferated (i.e. diluted the dye) and lost CCR6 expression. However, stimulated cells treated with mitomycin C lost CCR6 but did not divide. Therefore we had already fulfilled this question, but with a more sensitive measure of proliferation than BrdU incorporation. We have amended the text to more clearly highlight these data (lines 257-262).

8. The data in the melanoma and EAE models suggest that CCR6 is not required for cellular trafficking, however CCR2 is essential. In the *S. pneumonia* infection model, it is unclear if CCR6 expression is required for protective $\gamma\delta$ T cell responses as only the effects with CCR2^{-/-} $\gamma\delta$ T cell transfer were assessed.

Experiments with *S. pneumoniae* utilised transfers of *in vitro* **activated** and expanded $\gamma\delta$ T17 cells. As we clearly demonstrate, activated $\gamma\delta$ T17 cells lack CCR6 expression (Fig 1). Accordingly, $\gamma\delta$ T17 cells expanded *in vitro*, and used in these experiments, lack functional CCR6 expression as expected (Fig 6b/c - empty virus-transduced cells). Thus, as the cells do not express CCR6, it is evident that in these experiments, CCR6 is not required for $\gamma\delta$ T17 cell trafficking and/or protection.

Reviewer #2 (Remarks to the Author):gd T

This study by McKenzie et al defines a differential role for CCR2 and CCR6 in gdT17 homing to inflamed tissues and the dermis, respectively. Upon activation, the gdT17 cells downregulate CCR6, releasing them to migrate to inflamed sites via CCR2. I found this study most interesting and it clearly represents a robust body of work. The model proposed by the authors may appeal to a broader audience particularly since this type of "release" mechanism appears quite novel.

I have some minor comments on this study:

1. Fig 2 - Is there an effect of CCR6, CCR2 and CCR2.CCR6 KO on tumor growth/EAE scores?

We have included data demonstrating the effect of *Ccr6*, *Ccr2* or *Ccr2 Ccr6*

deficiency upon EAE scores and tumor weights at d7 post-challenge (Fig S3). As there are clear differences, we highlight the necessity of our co-transfer experiments to assess cell-intrinsic recruitment of $\gamma\delta$ T17 cells to these inflammatory sites (lines 146-148).

Figure S3 a) Average mass (per mouse) of B16 melanomas 7d post-challenge in WT (n=19), *Ccr6*^{-/-} (n=18), *Ccr2*^{-/-} (n=19) and *Ccr2*^{-/-}*Ccr6*^{-/-} mice (n=13). **b)** Clinical disease scores of WT, *Ccr6*^{-/-}, *Ccr2*^{-/-} and *Ccr2*^{-/-}*Ccr6*^{-/-} mice given EAE (n=17/group). Mean±SD. a-b) Pooled from two experiments. a) One-way ANOVA with Bonferroni's multiple comparisons test.

2. Fig 2 vs Fig.4. It is not clear why in some cases the authors use CD3-bright gdT17 cells whereas others CD3-lo.

The use of these gating strategies has been previously published and we have clarified these approaches in the results section (lines 93-95, 194-195). In short:

- CD3-bright is a phenotype used to distinguish V γ 6⁺ (CD3-bright) $\gamma\delta$ T17 cells from V γ 4⁺ ('conventional' CD3 staining) $\gamma\delta$ T17 cells as is published (Paget *et al.* 2015 Immunol Cell Biol, ref #25). We use this as a complementary strategy to V γ 4 staining to show that V γ 4⁺ and V γ 6⁺ $\gamma\delta$ T17 cells share trafficking characteristics.
- CD3^{lo} is a phenotype used to distinguish dermal $\gamma\delta$ T cells (CD3^{lo}TCR- $\gamma\delta$ ^{lo} i.e. $\gamma\delta$ T^{lo}) from contaminating dendritic epidermal T cells (DETCs, CD3^{hi}TCR- $\gamma\delta$ ^{hi}) in dermal cell preparations (Gray *et al.* 2011 J. Immunol, ref #22). We only refer to $\gamma\delta$ T^{lo} cells when describing dermal $\gamma\delta$ T cells, which are essentially all $\gamma\delta$ T17 cells (Fig S5a)

3. Whilst the importance of human gdT17 cells is less clear, it is an obvious question that may be worth discussing. Eg do any human gdT cells express these chemokine receptors and are they modulated by stimulation?

We agree and have now included a discussion of this as yet unexplored area in the revised manuscript (lines 373-384).

Reviewer #3 (Remarks to the Author):gd17, cytokine

Authors Duncan R.McKenzie et al. describe "IL-17-producing $\gamma\delta$ T cells switch migratory pattern between resting and activated states". Authors found that

“naturally occurring” IL-17A-eYFP+ $\gamma\delta$ T cells coexpress CCR2 and CCR6 in thymus, spleen and LN, irrespective of V γ 4 or V γ 6 expression, and the CCR6-CCR2+ IL-17A-eYFP+ $\gamma\delta$ T cells were increased in non-lymphoid organs such as , lung , PP and si LPL of naive mice and in the LN and CNS after induction of inflammation such as EAE, Melanoma and S.pneumoniae infection. CCR6 are down regulated in Brdu+ IL-17A-eYFP+ $\gamma\delta$ T cells during inflammation or after in vitro stimulation, presumably via BATF and IRF4 induction. This is well-performed paper and their findings are interesting and informative for localization of IL-17-producing $\gamma\delta$ T cells in inflamed tissues. My comments are as follows:

General comment.

Notable finding in this paper are in vivo significance of down regulation of CCR6 activated $\gamma\delta$ T cells and its molecular mechanism. However, authors show only difference in Fig 6, in which CCR6 transduced IL-17A-eYFP+ $\gamma\delta$ T cells migrated away from tumor. Authors also should do this experiment in EAE and S. pneumoniae infection. For molecular mechanism, author show only descriptive data that BATFKO or IRF-4 KO IL-17A-eYFP+ $\gamma\delta$ T cells fail to downregulate CCR6 only partly after in vitro stimulation with IL-23 and IL1beta.

Replies to these general comments are included below.

Specific comments

#1. Figure 2: Score of EAE may be ameliorated in CCR2-/- mice. Tumor size may be enlarged in CCR2-/- mice. Authors show the progress in diseases in CCR2-/- mice There are only small difference in migration in inflamed sites between activated CCR2-/- and CCR2+/+ IL-17A-eYFP+ $\gamma\delta$ T cells. FACS profile in EAE should be shown in transfer experiments.

We have included EAE scores and tumor sizes (Fig S3a) in *Ccr6*, *Ccr2* and *Ccr2* *Ccr6* deficient mice (also see point 1, reviewer 2). As CNS inflammation is reduced, while tumor size increased, in *Ccr2* deficient mice, we highlight the importance and novelty of our co-transfer experiments in Fig 2i-j (lines 146-148). We now show the flow cytometry for EAE transfer experiments (Fig 2j).

#2. Figure 5. Author show only descriptive data that BATFKO or IRF-4 KO IL-17A-eYFP+ $\gamma\delta$ T cells fail to down-regulate CCR6 after in vitro stimulation with IL-23 and IL1b. BATF is basic leucine zipper protein that dimerizes with JUN proteins to the transcriptional activity of functionally important genes in lymphocytes for functional maturation and migration of effector T cells. How BATF or IRF4 down-regulate CCR6?

We now include data from our own and others' ChIP-seq datasets identifying binding of both IRF4 and BATF to a conserved site in the *Ccr6* promoter in two murine T cell subsets (Fig S7b, lines 254-257). Therefore it is likely that IRF4 and BATF cooperate (as is reported, Ciofani *et al.* 2012 Cell) to repress *Ccr6* expression in $\gamma\delta$ T17 cells. We feel that further investigation of this point surpasses the scope of the manuscript, which concerns migration rather than molecular interactions. We believe that the ‘descriptive’ data included in the original submission, i.e. that IRF4 and BATF are required for CCR6 downregulation, are the most relevant for the biological function of $\gamma\delta$ T17 cells and thus the message of the paper.

Figure S7 b) IRF4 and BATF ChIP-Seq data at the *Ccr6* locus from CD8⁺ T cells and Th17 cells from published datasets. Line indicates binding site consistently detected for both transcription factors in both cell types. Datasets are from Kurachi *et al.* 2014 (BATF CD8⁺ T cells), Man *et al.* 2013 (IRF4 CD8⁺ T cells) and Ciofani *et al.* 2012 (IRF4/BATF Th17 cells) as referenced in results section.

Author conclude that down-regulation of CCR6 occurs independent to cell proliferation. Dose Mytomycin C inhibit cell proliferation sufficiently?

The submitted manuscript includes the use of cell proliferation eFluor670 dye to demonstrate that mitomycin C treatment completely blocked $\gamma\delta$ T17 cell proliferation upon stimulation (Fig 5f). We reiterate this point in the text (line 259-261).

Do BATFKO or IRF-4KO $\gamma\delta$ T cells express IL23R, IL1R, or IL-17A normally?

We have analysed resting $\gamma\delta$ T17 cells from WT, *Irf4*^{-/-} and *Batf*^{-/-} mice. We thank the reviewer for the important suggestion to determine whether $\gamma\delta$ T17 cells in these mice express normal amounts of IL-23R and IL-1R1 as this could impact upon their activation. As is shown below and now included in the manuscript (Fig S7a, lines 251-253), *Irf4*^{-/-} and *Batf*^{-/-} $\gamma\delta$ T17 cells express similar levels of both receptors to WT cells. Therefore we maintain our conclusion that reduced CCR6 repression and proliferation of these cells in response to IL-23 and IL-1 β stimulation is due to acute IRF4 and BATF-mediated signaling, and not due to IRF4 and BATF-driven surface expression of the appropriate stimulatory receptors.

Figure S7 a) Representative flow cytometry of IL-23R and IL-1R1 expression by splenic CD44^{hi}CD27⁻ $\gamma\delta$ T cells from WT, *Irf4*^{-/-} and *Batf*^{-/-} mice (n=3/group).

The submitted manuscript demonstrated that $\gamma\delta$ T17 cells are less frequent in *Irf4*^{-/-}

mice (Fig S6c, formerly Fig S4c). We show below that $\gamma\delta$ T17 cells in all strains express similar amounts of IL-17A on a per cell basis (based upon gMFI). However, this does not affect our analysis of CCR6 downregulation *in vitro*, as all such analyses (Fig 5e) were pre-gated on IL-17⁺ $\gamma\delta$ T cells. Therefore differential frequency of IL-17⁺ cells does not change our conclusion that IRF4 and BATF drive CCR6 downregulation upon activation.

IL-17A gMFI within splenic CD3⁺TCR- $\gamma\delta$ ⁺CD44^{hi}IL-17A⁺ $\gamma\delta$ T17 cells from WT, *Irf4*^{-/-} and *Batf*^{-/-} mice (n=3/group). Representative of two experiments.

BATF regulates localization of $\alpha\beta$ T cells in intestinal niches via directing expression of CD103 and CCR9 (J.E.M. 210:475-489, 2010). BALT KO mice may show an abnormal localization of $\gamma\delta$ T cells.

While this may be the case, we are investigating with the role of CCR6 and CCR2 in $\gamma\delta$ T17 cell biology. We consider that pursuing the role of BATF in $\gamma\delta$ T cell localization regarding CCR9 is beyond the scope of this manuscript.

CCR6 promoter/enhancer assay may clarify the direct effect of these molecules on cytokines-induced down-regulation.

This is addressed above.

#3 Figure 6: This is one of the most important figures in this paper, showing *in vivo* a significance of down regulation of CCR6 on $\gamma\delta$ T cells for preventing sequestration into uninflamed dermis from inflamed tissues. However, authors show only minor difference in Fig 6, in which CCR6-transduced IL-17A-eYFP+ $\gamma\delta$ T cells migrated away from tumor. Author should show that the CCR6-transduced IL-17A-eYFP+ $\gamma\delta$ T cells $\gamma\delta$ T cell in the dermis Authors also should do this experiment in EAE and *S. pneumoniae* infection and the $\gamma\delta$ T cell in the dermis

The experiment in the submitted manuscript demonstrates that while *Ccr6*-tg $\gamma\delta$ T17 cells were deficient at homing to B16 melanomas, they instead homed to uninflamed dermis (Fig 6d). As suggested, we have now performed this experiment in EAE and *S. pneumoniae* as requested, and obtained complementary results: *Ccr6*-tg $\gamma\delta$ T17 cells were less able to home to the inflamed CNS (EAE) and nasal passage (*S. pneumoniae* infection) than control-transduced $\gamma\delta$ T17 cells. Moreover, *Ccr6*-tg $\gamma\delta$ T17 cells preferentially homed instead to uninflamed dermis during *S. pneumoniae* infection. Note that it was not possible to obtain sufficient uninflamed skin during EAE to assess dermal homing in this model (due to subcutaneous CFA injection in both hind

flanks precluding the harvest of back skin). These experiments are included in Fig 6e-f (lines 291-296). These new data support our previous conclusion: maintained CCR6 expression diverts activated $\gamma\delta$ T17 cells into uninflamed skin and thus away from inflamed tissue. We believe that these data significantly strengthen the manuscript and we thank the reviewer for this suggestion.

e-f) *In vitro*-expanded $\gamma\delta$ T17 cells from WT or F₁ mice were transduced with empty pMIG or pMIG-*Ccr6*, respectively. Equal numbers of mixed GFP⁺ cells were transferred i.v. into Ly5.1 mice either e) 24 hr post-infection with *S. pneumoniae* (n=4) or f) at EAE onset (n=3) and organs were analysed 48 hr later. Ratio of recovered WT to F₁ $\gamma\delta$ T17 cells within transduced (GFP⁺) and untransduced (GFP⁻) populations, normalized to input values. e-f) Paired two-tailed Student's *t*-test.

Minor comments

#1 Introduction: many references are misscited. Authors should cite more original papers.

We have thoroughly checked all references and believe that all are correctly cited. The sole review cited in the original manuscript has been supplemented with a number of original primary papers in the revised manuscript (line 192).

#2 Figure 4: What is the role of CCR6 on IL-17A-eYFP+ $\gamma\delta$ T cells except migration to dermis? Is there any difference in localization of CCR6^{-/-} $\gamma\delta$ T cells in other organs.

Of all tissues examined, the only to demonstrate a reduction in $\gamma\delta$ T17 cells in *Ccr6*^{-/-} mice was the dermis (Fig 4a and Fig S5b, formerly Fig S3b). We found a greater number of $\gamma\delta$ T17 cells in the peritoneal cavity of *Ccr6*^{-/-} mice therefore it is likely that $\gamma\delta$ T17 cells unable to enter the dermis pool in this location instead.

RESPONSE TO REVIEWERS' FINAL COMMENTS

Reviewer #1 (Remarks to the Author):

The authors have addressed all my comments either by arguing them or with additional data. While I don't accept all the arguments, the manuscript is improved and represents a nice body of work.

We thank the reviewer for their effort and for their excellent suggestions to improve our manuscript.

Reviewer #2 (Remarks to the Author):

I thank the authors for their rebuttal. I am satisfied with this response and have no further comments.

We thank the reviewer for their effort and for their excellent suggestions to improve our manuscript.

Reviewer #3 (Remarks to the Author):

Authors responded satisfactorily to the reviewer's comments and suggestions.

We thank the reviewer for their effort and for their excellent suggestions to improve our manuscript.